The inorganic pyrophosphatases of microorganisms: a structural and functional review

García-Contreras Rodolfo 1
de la Mora Javier 2
Mora-Montes Héctor Manuel 3
Martínez-Álvarez José A. 3
Vicente-Gómez Marcos 3
Padilla-Vaca Felipe 3
Vargas-Maya Naurú Idalia ni.vargas@ugto.mx 3
Franco Bernardo bfranco@ugto.mx 3
1 Departamento de Microbiología, Facultad de Medicina, Universidad Nacional Autónoma de México , Mexico City , Mexico
2 Genética Molecular, Instituto de Fisiología Celular, Universidad Nacional Autónoma de México , Mexico City , Mexico
3 Departamento de Biología, División de Ciencias Naturales y Exactas, Universidad de Guanajuato , Guanajuato , Mexico
Wang Liang
Electronic publication date: 2024 Jun 24
Publication date: 2024
Volume: 12
Electronic Location ID: e17496
Received 2024 Jan 23; Accepted 2024 May 9
Copyright: ©2024 García-Contreras et al.
Copyright year: 2024
Copyright holder: García-Contreras et al.
License: This is an open access article distributed under the terms of the Creative Commons Attribution License, which permits unrestricted use, distribution, reproduction and adaptation in any medium and for any purpose provided that it is properly attributed. For attribution, the original author(s), title, publication source (PeerJ) and either DOI or URL of the article must be cited.
License URL: https://creativecommons.org/licenses/by/4.0/

Keywords: Cytoplasmic pyrophosphatase, Membrane-bound pyrophosphatase, Catalysis, Bacteria, Biotechnological applications

Funding: The authors received no funding for this work.

==============================
Pyrophosphatases (PPases) are enzymes that catalyze the hydrolysis of pyrophosphate (PPi), a byproduct of the synthesis and degradation of diverse biomolecules. The accumulation of PPi in the cell can result in cell death. Although the substrate is the same, there are variations in the catalysis and features of these enzymes. Two enzyme forms have been identified in bacteria: cytoplasmic or soluble pyrophosphatases and membrane-bound pyrophosphatases, which play major roles in cell bioenergetics. In eukaryotic cells, cytoplasmic enzymes are the predominant form of PPases (c-PPases), while membrane enzymes (m-PPases) are found only in protists and plants. The study of bacterial cytoplasmic and membrane-bound pyrophosphatases has slowed in recent years. These enzymes are central to cell metabolism and physiology since phospholipid and nucleic acid synthesis release important amounts of PPi that must be removed to allow biosynthesis to continue. In this review, two aims were pursued: first, to provide insight into the structural features of PPases known to date and that are well characterized, and to provide examples of enzymes with novel features. Second, the scientific community should continue studying these enzymes because they have many biotechnological applications. Additionally, in this review, we provide evidence that there are m-PPases present in fungi; to date, no examples have been characterized. Therefore, the diversity of PPase enzymes is still a fruitful field of research. Additionally, we focused on the roles of H+/Na+ pumps and m-PPases in cell bioenergetics. Finally, we provide some examples of the applications of these enzymes in molecular biology and biotechnology, especially in plants. This review is valuable for professionals in the biochemistry field of protein structure–function relationships and experts in other fields, such as chemistry, nanotechnology, and plant sciences.

Introduction

Pyrophosphatases (PPase; c-PPases for cytoplasmic PPase, EC 3.6.1.1) are enzymes that catalyze the hydrolysis of pyrophosphate (PPi) to organic phosphate (Pi). c-PPase has been demonstrated to be essential in Escherichia coli (Chen et al., 1990). In yeast, PPase conditional defects lead to massive cell death in fermenting cells, while in respiring cells, only cell cycle arrest occurs (Serrano-Bueno et al., 2013). The role of PPi in bioenergetics is to make biosynthetic reactions thermodynamically favorable when it is removed, specifically in reactions such as nucleic acid synthesis; the production of coenzymes, proteins, isoprenoids; and the activation of fatty acids (Kornberg, 1962; Lahti, 1983; Pace et al., 2011).

c-PPase is essential for cell viability, mainly being involved in DNA and phospholipid synthesis homeostasis. In contrast, the bacterial membrane-bound enzyme (m-PPase) is coupled to produce a transmembrane electrochemical potential of an ion, producing a proton pump enzyme that uses low-cost fuel (Hirono et al., 2007).

However, one key question has been focused on: what is the role of PPi? Two main views in this regard have been proposed. First, PPi was the ancient energy currency used by the first forms of life (Lipmann, 1965), further supported by the discovery of an m-PPase that generates a transmembrane electrochemical potential of an ion, either H+ or Na+ ions (Baltscheffsky et al., 1966). Second, alternatively, the production of PPi is a byproduct of all biosynthetic pathways that generate this molecule and is not used as an energy source (Wimmer, Kleinermanns & Martin, 2021); specifically, the growth of a cell is linked to the synthesis of PPi, while its hydrolysis is not (Wimmer, Kleinermanns & Martin, 2021).

Recently, a thorough analysis has shown that PPi is not used as an energy source, and there is no evidence of this phenomenon in 402 reactions of the universal biosynthetic core: none of the core reactions use PPi. Interestingly, 36% of phosphoanhydride hydrolyzing reactions generate PPi (Wimmer, Kleinermanns & Martin, 2021). Additionally, for the main body mass that is produced from the synthesis of DNA, RNA, and proteins, which is approximately 80% of the cell mass, all reactions generate PPi, while none of these core reactions use PPi as an energy source. The work by Wimmer, Kleinermanns & Martin (2021) also showed that PPi hydrolysis is an ancient mechanism that imparts the irreversibility of many biosynthetic reactions, suggesting that PPi was never involved as an energy currency and that perhaps PPi is a mechanistic molecule that pushes reactions toward synthesis and growth by exerting kinetic control over the reactions. Thus, the seminal work by Wimmer, Kleinermanns & Martin (2021) supports the role that Kornberg (1962) proposed for PPi in 1962 as the driving force in many instances to prevent reversibility of the biochemical reactions leading to synthesis. Additionally, regarding the core metabolic role of the last universal common ancestor, Wimmer, Kleinermanns & Martin (2021) suggested that PPi hydrolysis functions as a pawl in a ratchet’s pawl mechanism, driving the irreversibility of reactions, which is strongly supported by the continued degradation of PPi in cells where biosynthetic pathways are halted. Additionally, Danchin, Dondon & Daniel (1984) reported that 2-ketobutyrate is an alarmon that reduces the levels of phosphorylated substrates such as fructose-1,6-diphosphate, glucose 6-phosphate, and acetyl-CoA, which is related to reduced biosynthetic reactions; this is also true for m-PPase, which needs an ion gradient for PPi synthesis that depends on ATP (Baltscheffsky et al., 1966; Baltscheffsky, 1967). An m-PPase requiring Na+ ions was discovered in Thermotoga maritima (Belogurov et al., 2005), which can function as a proton pump at low sodium concentrations (Luoto et al., 2013).

One interesting exception is the synthetic reaction in examples of photosynthetic bacteria, such as Rhodospirillum rubrum. In this bacterium, the photosynthetic machinery is compartmentalized in chromatophores. m-PPase (EC 7.1.3.1) is present in these structures and requires a transmembrane electrochemical potential of an ion generated by photosynthetic reactions (Baltscheffsky & Baltscheffsky, 1995). García-Contreras, Celis & Romero (2004) demonstrated that in R. rubrum, in the absence of m-PPase under low-energy conditions, the pH gradient collapsed in a mutant strain lacking membrane PPase, and growth was halted under low light intensity (2 W/m2). Nevertheless, in this mutant strain, c-PPase remained active.

In this review, we aimed to provide an overview of c- and m-PPases to fuel the research in this field and provide insight into organisms showing divergence in mainstream research on these hydrolytic enzymes. Additionally, we aimed to elucidate some aspects related to the applications of PPases in both molecular biology and biotechnology. Complementary to this review, we conducted bioinformatic analysis using UniProt and structural modeling comparisons to highlight the fields that remain to be elucidated regarding the function and diversity of these enzymes.

General Aspects of PPases

Overall, the catalytic activity of PPases involves the hydrolysis of PPi into two phosphate ions, and in most organisms, this reaction is conducted by cytoplasmic enzymes (Heinonen, 2001; Baykov et al., 2013; Baykov et al., 2017). The reaction can be represented as follows:

Inorganic pyrophosphate (PPi) + H2O ⟶PPase 2 phosphate ions (Pi)

This reaction is energetically favorable, as the cleavage of the high-energy pyrophosphate bond releases energy; in the case of c-PPases, this energy is released in the form of heat, and in the m-PPases, it is released in the form of H+ or Na+ transport across membranes (present in plants and some archaea, bacteria, and protists) (Baykov et al., 2017). Hydrolysis follows a stepwise mechanism that involves the binding of Mg-PPi and subsequent hydrolysis of the enzyme, followed by the sequential removal of two Pi molecules, HPO42− and/or MgHPO4 (Baykov et al., 2017).

The catalytic reaction steps for PPases are as follows (Baykov et al., 2017):

     k1    k2    k3    k4

EM3+MPP ⇆EM4PP ⇆EM4P2 ⇆EM3+MP ⇆EM3+MP+P

     k−1    k−2   k−3    k−4

In the catalytic reaction, M denotes the presence of a metal cofactor (Mn2+, Co2+, or Mg2+) in the protein structure or the metal bound to the substrate, P refers to phosphate, and PPi refers to pyrophosphate (Baykov et al., 2017). The rate of each k constant is available in the work by Baykov et al. (2017). The catalytic rate constants are greater for family II PPases in the substrate binding step, while the PPi hydrolysis proceeds faster than product removal in both families (Baykov et al., 2017).

The mechanistic details of the catalysis of family I enzymes have been described for yeast PPase. Using high-resolution crystallography, Heikinheimo et al. (2001) described the role of metal ions in the active site, showing that their role is to form metal-coordinated anions that attack the phosphate monoester dianions via an associative mechanism. Furthermore, the authors unambiguously demonstrated that hydrolysis is achieved by attaching a water molecule directly to PPi with a pK a lowered by two metal ions and Asp117. The reaction with water molecules is achieved by the formation of a low-barrier hydrogen bond between Asp117 (orange in Fig. 1A), and the water molecule, which is located in the active site and is pushed by Trp100 hydrophobicity and attracted by Asn116 (deep purple in Fig. 1A) (Heikinheimo et al., 2001). The active site residues are shown in Fig. 1A. The mechanistic role of water and Asp117 has also been supported by site-directed mutagenesis and the formation of the intermediary nucleophile (Pohjanjoki et al., 2001). Thus, forming a low-barrier hydrogen bond, along with the metal ion, seems to provide the enzyme with catalytic power. Additionally, glycine residues are associated with protein stability and the establishment of β-turns in the structure (Moiseev, Rodina & Avaeva, 2005). In E. coli, Moiseev, Rodina & Avaeva (2005) studied glycine residues flanking the active site and demonstrated that Gly100 and Gly147 are involved in the correct folding of the E. coli PPase; mutations of these residues result in major accessibility to denaturants, which results in subunit dissociation (Moiseev, Rodina & Avaeva, 2005). In the yeast PPase, the glycine residues are located in a disordered region, similar to the reported structure in E. coli (Fig. 1A). The role of the glycine residues remains to be shown by point mutations in yeast PPase.

Figure 1 Structure of family I and II c-PPases and the active site residues.

(A) shows representative examples of family I c-PPases. The yeast structure was used to exemplify the fold and position of the catalytic residues (PDB: 1E9G, Heikinheimo et al., 2001). Orange indicates the residues Asp115, Asp152, and Asp120. The Tyr93 residue is indicated in green. In the blue Lys56 residue, PPi is shown as thick red sticks, and the Mg2 + ion is shown in light blue. (B) shows an example of a family II PPase. B. subtilis PPase was used as an example (PDB: 1K23, Ahn et al., 2001), and only one subunit is shown (extracted using UCSF Chimera software; https://www.cgl.ucsf.edu/chimera/). Catalytic residues are indicated in red (residues Arg295 and Lys296 belong to the C-terminal conserved signature SRKKQ). The Mn2 + ions (as dots) are indicated in light blue.

Using high-resolution crystallography may provide further insight into the mechanistic transitions observed in other PPase enzymes at the active site by using the native enzyme and inhibiting it with specific molecules. Additionally, the native conformation of cytosolic PPases depends on the family to which they belong. Bacterial family I enzymes have been found to form homohexamers (the native form of the enzyme), homotrimers, or homodimers, depending on specific mutations in the protein (Josse & Wong, 1971; Cooperman, Baykov & Lahti, 1992). For family II, the enzymes studied to date have been shown to form homodimers, except for enzymes bearing a regulatory domain (see the following sections) (Aravind & Koonin, 1998).

Nevertheless, some enzymes must be studied in detail since they contain novel domains. No crystallographic data or extensive mutagenesis analysis has characterized the roles of these domains in these enzymes, which will be discussed further in this review.

However, m-PPases are part of the PPi degrading machinery and are capable of pumping either H+ or Na+ ions. m-PPases, simple energy-supplying enzymes, have no known homologs to ion pumps, suggesting a unique ion transport coupling mechanism (Baykov et al., 2022). The origin of H+ PPases has been analyzed and they are suggested to have been derived from Na+/H+ cotransporter m-PPases, the most ancestral form of the enzyme (Luoto et al., 2011). Luoto et al. (2011) used detailed phylogenetic reconstruction to identify m-PPase lineages that independently acquired Na+ and H+ cotransporter functions and then evaluated their activity in the presence of Na+ ions at different concentrations, suggesting that m-PPase exhibits plasticity in ion transport. The two types of m-PPases identified belong to Clostridium lentocellum and Clostridium leptum, which lost their H+-transport efficiency at high [Na+], and the authors classified these enzymes as Na+-regulated Na+ and H+-PPases. In contrast, the m-PPases from Akkermansia muciniphila, Bacteroides vulgatus, and Prevotella oralis maintained strong H+ transport activity at all tested Na+ concentrations and are therefore true Na+ and H+ m-PPases; this provides experimental evidence linking the two functionally distinct Na+ and H+-PPase subtypes, and along with phylogenetic analysis, indicates that Na+ m-PPase is the most ancient form of the enzyme. Thus, the increase in H+ pump activity occurred later in evolution.

Nevertheless, the mechanistic function of these enzymes has been elucidated. The reaction mechanism of m-PPase enzymes can be summarized in a few key steps:

Substrate binding: The enzyme first binds to its substrate, PPi; it is important to indicate that the substrate for these enzymes is Mg2PPi, so once hydrolyzed, the phosphate is released as mono-magnesium complexes (Baykov et al., 2022).

Catalytic site activation: The enzyme contains a catalytic site with specific amino acid residues that facilitate the hydrolysis of PPi. These residues include Asp for the binding of metal ions (negative charge) and positive residues that participate in substrate binding (Holmes, Kalli & Goldman, 2019) by coordinating Mg2+ and water (Holmes, Kalli & Goldman, 2019), except for the m-PPase of R. rubrum, for which Glu202 is essential for substrate binding (Malinen et al., 2004).

Hydrolysis of PPi in m-PPases: The catalytic site activates a water molecule, turning it into a hydroxide ion (OH−) through the action of nearby amino acid residues. This hydroxide ion then attacks the phosphoanhydride bond in PPi, causing it to break.

Formation of phosphate ions: As a result of hydroxide ion attack, the phosphoanhydride bond is cleaved, and two phosphate ions (Pi) are released. These phosphate ions are typically in the form of HPO42−.

Product release: The two phosphate ions are released from the enzyme’s active site into the surrounding environment, where the cell can utilize them for various cellular processes, such as ATP synthesis or nucleotide biosynthesis. In this stage, the water-derived proton exits the channel on the protein if the transport is coupled with hydrolysis or reprotonates the Glu residue in the gate of the protein, which is composed of semiconserved nonpolar residues (Holmes, Kalli & Goldman, 2019).

Enzyme reset: After the reaction is complete, the enzyme’s catalytic site is reset, and the enzyme is ready to bind and hydrolyze another PPi molecule. The enzymes form dimers, and hydrolysis is theorized to occur in one subunit at a time (Holmes, Kalli & Goldman, 2019; Baykov et al., 2022). Figure 2A shows the regions of the Vigna radiata enzyme. Importantly, the cytoplasmic side with the hydrolytic center is indicated, as is the coupling funnel for the proton or sodium pass, the two gates needed for ion pumping regulation, and the exit channel. Figure 2 shows the positions of key residues in the hydrolytic center.

Figure 2 Structural aspects of m-PPases and the catalytic residues.

The PDB file 4A01 from V. radiata is shown, and the catalytic residues are indicated. The overall structure is shown in (A), and the relative position of the membrane is displayed. The arrow indicates the H+ or Na+ translocation flow in these enzymes, and the notes indicate the known regions in the catalytic center of a single subunit. In (B), the upper part of the protein is displayed, where red indicates the position of PPi, green indicates bound Mg2 + ions, and purple indicates the K+ ion position. In (C), a close-up image of the catalytic region of a monomer is shown. In (D), the positions of the catalytic residues are indicated with letters and numbers. Dotted K+ ions are indicated in purple, and Mg2 + ions are shown dotted in cyan. The weblogo displays the motif associated with K+ binding in selected examples of K+-regulated m-PPases. The bound PPi is indicated in magenta.

The m-PPases characterized to date are hypothesized to form dimers in the membrane (López-Marqués et al., 2005), which is in accordance with a recent study that demonstrated that they tend to undergo an asymmetric catalytic hydrolysis mechanism and thus show allostery in catalytic activity. In Thermotoga maritima m-PPase, when one subunit binds the substrate, the other monomer undergoes a conformational change, modifying its affinity for PPi. When the second monomer binds PPi, the first monomer can hydrolyze PPi, but the second monomer pumps H+ (Vidilaseris et al., 2019).

The binding site for K+ is indicated in Fig. 2D. The catalytic residues involved in substrate binding and K+ ion binding in the m-PPases sequences corresponds to a conserved signature shown in a weblogo of 18 m-PPases K+-sensitive regulated enzymes (Belogurov & Lahti, 2002; Serrano et al., 2007).

The activity of these enzymes may or may not be stimulated by K+ ions. K+-dependent enzymes are rare in prokaryotes (Baykov et al., 2013) but are more abundant in plants and protists (Maeshima, 2000; Drozdowicz & Rea, 2001; Pérez-Castiñeira et al., 2002a; Docampo & Moreno, 2008; Baykov et al., 2013). Additionally, these enzymes have not been identified to date in fungi and animals except for the yeast Saccharomyces carlsbergensis (Saccharomyces pastorianus) (Lichko & Okorokov, 1991). GenBank and UniProt searches failed to identify a coding sequence consistent with an m-PPase.

In the following sections, we review the available information regarding the mechanistic aspects of different pyrophosphatases and how they may involve an evolutionary hindrance that has prevented these enzymes from evolving or diverting from their main function, except for the m-PPases that have coupled the hydrolysis of PPi with proton or sodium pump activity. Additionally, with the rapid development of deep learning algorithms, we can now grasp at the protein universe scale the folding of proteins and uncover novel protein families as well as novel protein folding in the available genomes (Durairaj et al., 2023; Barrio-Hernandez et al., 2023). Here, we propose that the two families of cytoplasmic PPases diverged from a common ancestor. Some proteins have domains with regulatory roles that may have led to more complex regulatory networks than those in the c-PPases without these domains.

Cytoplasmic PPases, Families and Structural Features

As discussed previously, the catalytic mechanism and other relevant features of family I PPases have been extensively analyzed. In the “PPase diversity” section, we discuss the diversity of cytoplasmic enzymes and aspects that have not been studied in detail.

The distribution of c-PPases in all kingdoms of life suggests that there are organisms with a specific type of c-PPase (either family I or II), and this is evidently related to the environmental conditions for specific taxa and the encoded enzyme. For example, Vibrio cholera encodes two cytoplasmic PPases that seemed to cooperate; however, in V. cholerae cells, only the PPase from family I appears to be active (Salminen et al., 2006; Baykov et al., 2017). The presence of the two PPases also suggested that a third family may exist that is present in Haemophilus influenzae, Mannheimia succiniciproducens and Nitrosococcus oceani, which cannot be classified into the two known families (Salminen et al., 2006). Salminen et al. (2006) proposed that family I PPases belong to a monophyletic group. However, the bioinformatic data analyzed for this review shown in Fig. 3C and Fig. 4 suggest that family I comprises two groups, one phylogenetically associated only with bacteria and another widespread in protists, algae, and higher organisms. In contrast, family II enzymes are more closely related and phylogenetically grouped in an evolutionary sense (Fig. 4), even enzymes that present additional regulatory domains (see below).

Figure 3 The family I and II PPases have conserved sequences but can be differentiated into three subfamilies.

(A) sequence alignment (using Clustal Omega) visualized in Alignment Viewer (https://alignmentviewer.org/). The purple rectangle encloses the family II PPases. Additionally, this panel shows the two-color schemes used in the MSA view and charged or hydrophobic residues (upper and lower figures, respectively); in red, positively charged residues; in blue, negatively charged; and in white, hydrophobic residues. The figure also shows distinctive conserved sequences exclusively found in family II PPases (*) and the SRKKQ C-terminal signature that is conserved in both families (**); black arrows indicate the catalytic residues in the family I PPases. (B) shows the pairwise identity 2D map of the whole alignment, indicating the line corresponding to a specific PPase. In these cases, the three clusters observed correspond to family II in the upper cluster and the two distinctive family I clusters. In (C) UMAP (McInnes et al., 2018) was used to estimate the spatial distribution of the aligned sequences. Again, three clusters are found in 500 epoch iterations. Families are indicated by brackets (family I) and purple lines (family II).

Figure 4 Sequence diversity and structural features of c-PPases.

(A) is a phylogenetic reconstruction of examples of family I and II PPases. Evolutionary analysis by the maximum likelihood method was inferred using the maximum likelihood method and the JTT matrix-based model (Jones, Taylor & Thornton, 1992). The tree with the highest log likelihood (−18,445.28) is shown. The percentage of trees in which the associated taxa clustered together is shown below the branches. Initial tree(s) for the heuristic search were obtained by applying the neighbor-joining method to a matrix of pairwise distances estimated using the JTT model. The tree is drawn to scale, with branch lengths measured as the number of substitutions per site. This analysis involved 34 amino acid sequences. There were 760 positions in the final dataset. Evolutionary analyses were conducted in MEGA11 (Tamura, Stecher & Kumar, 2021). The branches were manually colored to indicate the following: purple, protist; light green, algae; dark green, plants; yellow, invertebrates; and blue, vertebrates. Bacteria are not colored. The brackets indicate the clusters for family I and family II.

Another pathogen from which cytoplasmic PPase has been studied, Helicobacter pylori, lacks an m-PPase. However, the cytoplasmic enzyme is homodimeric and sensitive to NaF (Oliva et al., 2000). This enzyme may represent a novel target for treatment. When nonconserved Cys16 is substituted with serine, it loses 50% of its activity, has reduced sensitivity to reducing agents, and experiences a decline in thermostability (Lee et al., 2007). The effect on the active site is related to the environment surrounding Cys16 in the active site, with the valine residues forming a hydrophobic cluster.

PPase diversity

To date, two main c-PPase families have been identified: family I and family II. In contrast, family III is less studied, has an enzymatic mechanism that involves the binding of magnesium-linked PPi, requires four kinetic constants for the binding and nucleophilic attack of the substrate, and involves haloalkane dehalogenases (see below) (Baykov et al., 1999). Using bioinformatic tools, we show in Fig. 3 examples of the sequence diversity in family I and II representatives. A third family has been proposed, but little information is available on these enzymes (see below). In Fig. 3A, the sequence diversity reflects that the enzymes have incorporated additional sequences, but the main catalytic residues remain the same. Family II, for instance, has a conserved SRKKQ signature at the C-terminal end. However, both families have conserved estimated physicochemical features in most full-length sequences (Fig. 3A). Curiously, Giardia intestinalis encodes a family II protein with regulatory domains that have been identified in bacteria, such as Clostridium perfringens (Tuominen et al., 2010), and has been proposed as a target for inhibition by AMP or ADP or activation by ATP or diadenosine polyphosphates (Anashkin et al., 2015), suggesting that the regulatory role of the CBS domain links PPi hydrolysis with the capacity for biosynthesis in bacteria. Family II proteins tend to cluster together even with extra regulatory domains. At the same time, family I seems to diverge more between organisms distant from bacteria and early divergent protists (Figs. 4B and 4C).

As shown in Fig. 3, in a bioinformatic comparison of a sample of cytoplasmic PPases from different organisms, families I and II displayed low sequence and structural homology. Although the overall catalytic mechanism is similar, regulatory differences have been found (Baykov et al., 2017; Sarmina, Peña Segura & Celis, 2017). Figure 3 shows that the regions conserved between the two families of cytoplasmic PPases are limited. The regions conserved between family I and family II members differed strongly, except that the estimated physicochemical properties were similar in most regions of the protein sequence; this can be seen more clearly when comparing a 2D pairwise alignment map. Family II PPases cluster together, and two clear clusters of family I proteins can be identified (bioinformatic analysis is shown in Fig. 3B; UMPA analysis is shown in Fig. 3C).

In Fig. 4, the phylogenetic distribution analyzed for this review of PPases seems to cluster in terms of their source. Bacterial family I PPases are more distant from their eukaryotic counterparts, except for Chlamydia trachomatis PPase (UniProt B0B8Z8), which is located as a common ancestor for two branches of bacterial enzymes and a branch of protists (Chlamydomonas reinhardtii and Trichomonas vaginalis). The other branch that clusters together is for amoebozoan and fungal PPases. The distribution of invertebrate and vertebrate enzymes is consistent with the clustering shown in Figs. 3B and 3C. Family II enzymes are clustered together, and surprisingly, G. an intestinalis PPase is a family II enzyme (UniProt V6TM56) that contains a regulatory domain (see below).

The family I PPases have strong similarities. Figure 5A shows that the extensively studied yeast PPase (PDB ID: 1E9G) strongly resembles the modeled enzymes from Aspergillus nidulans and Candida albicans. However, Drosophila and Caenorhabditis elegans, in which no determined structure exists, both contain a predicted disordered N-terminal end (Fig. 5B). Nevertheless, the active site region is conserved compared to that of yeast PPase.

Figure 5 Structural features of family I enzymes and the evolutionary conservation of the catalytic folds.

(A) comparison of A. nidulans, S. cervisiae and C. albicans enzymes, showing that the examples shown here are highly conserved in comparison with the crystal structure of the S. cerevisiae enzyme (PDB 1E9G, RMSD 0.91, average TM-score = 0.74067). However, in (B), the comparison of the S. cerevisiae enzyme with the Drosophila and C. elegans enzymes reveals N- and C-terminal domains with no predicted fold, which may indicate that the S. cerevisiae enzyme is involved in intracellular localization (RMSD = 2.07, average TM-score = 0.82980 for the comparison between Drosophila and C. elegans, and the comparison with the S. cerevisiae enzyme showed an RMSD = 2.26 and an average TM-score of 0.526). Previously, McLaughlin, Lindmark & Müller (1978a); McLaughlin, Lindmark & Müller (1978b) determined that pyrophosphatase activity was localized in the plasma membrane in E. histolytica. (C) shows that the unique PPase in E. histolytica (UniProt S0AZM4) has no predicted transmembrane domains (Protter). The expected model using AlphaFold2 is shown in (D), using a rainbow color scheme, and only a C-terminal helix is found to be different from that of other family I enzymes. Surface charge prediction revealed that this helix is negatively charged, which suggests that this enzyme is unlikely to be associated with any cell membrane.

The N-terminal extensions found in Drosophila and C. elegans enzymes led us to search the literature for studies on m-PPase activity in other organisms. We found that two reports suggested that Entamoeba histolytica had PPase activity in postnuclear homogenates with localization at the cell membrane and distinctive biochemical features, such as activity at pH 5.0 and no divalent cation requirement (McLaughlin, Lindmark & Müller, 1978a). Additionally, the authors discuss the possibility that the enzyme is enclosed in an organelle. In a follow-up study, McLaughlin, Lindmark & Müller (1978b) demonstrated that the enzyme is heat stable, which is consistent with other family I c-PPases and the temperature sensitivity observed in some family II examples (Klemme & Gest, 1971; Romero, García-Contreras & Celis, 2003; Celis et al., 2003). However, the presence of this enzyme in the membrane was intriguing.

In Fig. 5C, using the Protter tool, analysis of the single pyrophosphatase (244 residues) annotated in UniProt (in different strains S0AZM4, M2Q2R6, A0A5K1U1F2, M7W242, M3SAR5, N9V599, and C4M982) revealed a C-terminal helix that is not found in typical PPases and has no residues that suggest binding to the membrane (Fig. 5D). E. histolytica HM-1:IMSS-B (UniProt M3TGX3) encodes a classic enzyme (175 residues). Recently, an enzyme was found by mass spectrometry in the mitosome of this parasite to colocalize with mitochondrial chaperonin 60 (Cpn60) (Mi-ichi et al., 2009) and it may play a role in sulfate metabolism via physical interactions with sulfate activation enzymes, since in E. histolytica, the c-PPase is not fused to APS kinase, ATP-sulfurylase, or a recognizable domain that suggests its involvement in sulfate metabolism (Bradley et al., 2009; Mi-ichi et al., 2009). Additionally, a similar extension was observed for the Saccharomyces carlsbergensis (Saccharomyces pastorianus) PPase (Lichko & Okorokov, 1991). In Fig. S2, the modeled PPase from S. pastoris shows a similar helix, in this case in the N-terminal end, which could explain the limited interaction with the membrane. However, no hydrophobic residues are found in this protein since the predicted surface charge shows some neutral residues, but several positive residues are shown; perhaps it interacts with another protein that is associated with the membrane, resulting in the determination of enzymatic activity as being associated with a membrane. Further biochemical data are needed to clarify the c-PPase activity in these organisms.

Cytoplasmic PPases, as stated above, are divided into two families, and a recently discovered third family has a complex biochemistry. The classification depends on the sequences and structure that impact their features. The family I PPases are present in all kingdoms of life and use Mg2+ as their cofactor. Their structure is either dimeric or hexameric (Heikinheimo et al., 1996; Harutyunyan et al., 1997; Baykov et al., 2017). Family II PPases are relatively rare, are found mainly in bacteria, and are more active when Mn2+ is used as their substrate cofactor (Fabrichniy et al., 2004), as they contain either Mn or Co bound to the enzyme. The proposed family III PPases belong to the haloalkane dehalogenase family, are found in some bacteria, and are Ni 2+-dependent (Lee et al., 2009). To date, the only structure reported for this family is Bacteroides thetaiotaomicron BT2127 inorganic pyrophosphatase (UniProt Q8A5V9) (Huang et al., 2011).

Using Fold Seek, we found that family III enzymes are not annotated as pyrophosphatases, but rather, using a model from the Thermococcus onnurineus enzyme (UniProt B6YSF3, Fig. 6A) (Lee et al., 2009), most hits are pyrimidine 5′ nucleotidases, haloacid dehalogenases or probable hydrolases. Figure 6 shows that the structures of these enzymes are similar to those of the pyrimidine 5′ nucleotidase PynA from Streptococcus pneumoniae (Fig. 6B), YfnB from B. subtilis (PDB 3I76, Fig. 6C), and a hypothetical protein from Pyrococcus horikoshii (PDB 2OM6, Fig. 6D). The structural alignment shown in Fig. 7E indicates that all of these proteins are closely related. However, bearing a different multimeric structure, the hypothetical protein from Pyrococcus horikoshii is a dimer, while YnfB from B. subtilis is a trimeric protein (Fig. 6). More work is needed to characterize and differentiate these enzymes from other functional metabolic enzymes since the results obtained with Fold Seek suggest that other enzymes bear a similar fold and may behave as moonlighting enzymes in addition to having PPase activity.

Figure 6 Family III PPases display highly conserved folding.

(A) shows the predicted structure of Thermococcus onnurineus (UniProt B6YSF3). (B) shows the predicted structure of PynA, a pyrimidine 5′-nucleotidase from Streptococcus pneumoniae (UniProt Q8DPQ3). (C) shows the crystal structure of YfnB from B. subtilis (PDB 3I76). (D) shows the crystal structure of a probable phosphoserine phosphatase from Pyrococcus horikoshii OT3 (PDB 2OM6). The dashed lines show the folding of the second subunit, showing the presence of the active residues (lines). (E) compares the monomers of all family III examples (RMSD =1.92, TM-score = 0.85562). Alignment: Thermococcus onnurineus, Streptococcus pneumoniae, B. subtilis, Pyrococcus horikoshii.

Figure 7 Family II enzymes that contain cystathionine β-synthase (CBS) regulatory domains also exhibit folding diversity.

(A) alignment viewer image of protein sequence alignment of examples of enzymes from family II that either lack the CBS regulatory domain or contain the tandem CBS domains. The MSA view, residue property color scheme, DHH catalytic signature, and C-terminal SRKKQ signatures are shown. (B) shows examples of different microorganisms that encode family II enzymes. Table 1 shows the BLASTp results and accession numbers for those containing a CBS domain. The B. subtilis enzyme is shown as a reference since the enzyme structure has been determined experimentally (PDB 1K23B). The color scheme used is rainbow.

Family II enzymes

Cytoplasmic PPases are enzymes needed to maintain the biosynthetic nature of metabolic pathways. Although the overall functions of family I and II enzymes are the same, there are some differences in their specific mechanisms and structures (Sarmina, Peña Segura & Celis, 2017). The family I PPases are usually homohexameric (except in eukaryotic organisms for which they are dimeric) with a monomer with a molecular mass of ∼20 kDa. Family II PPases are usually homodimeric enzymes with larger subunits (∼34 kDa per subunit); the active site is located at the interface of the two subunits, and the active site contains a conserved signature of DHH residues placed into two domains (DHH and DHHA2) (bioinformatic comparison shown in Fig. 7B, B. subtilis enzyme) (Merckel et al., 2001; Sarmina, Peña Segura & Celis, 2017). Additionally, the structure depends on an open and closed state for catalysis, as shown for the B. subtilis enzyme (Ahn et al., 2001). In the case of B. subtilis and Streptococcus gordonii, the two domains are linked by a flexible hinge, and the active site is formed between the two subunits (Fig. 1B); the distance between catalytic residues depends on the binding of the substrate, and the metal bound to the enzyme forms an activated water molecule that results in nucleophilic attack on the PPi (Ahn et al., 2001). The two regions comprising the active site were demonstrated to be directly involved in the two charged residues in the conserved SRKKQ C-terminal signature in the B. subtilis enzyme, and Arg295 and Lys296 are directly involved in substrate binding (Shizawa et al., 2001). One exception in the conformation of the active site is the case of Staphylococcus aureus. The enzyme is in a closed conformation in the absence of substrate, bound PNP (imidodiphosphate), sulfate or chloride, or reaction products; this is due to the position in this enzyme of the SRKKQ motif, which is outside of the active site and is proposed to be mobile upon substrate binding (Gajadeera et al., 2015). After searching the literature, we were not able to find a follow-up study identifying the role of the SRKKQ signature in the S. aureus enzyme, suggesting that further research is needed to elucidate the mechanistic effect of a closed conformation in PPi hydrolysis and perhaps the role of this enzyme as a therapeutic target in multidrug-resistant strains.

Important discoveries regarding the differences between family I and II PPases in photosynthetic bacteria exist. First, the family II PPases of Cereibacter sphaeroides (Rhodobacter sphaeroides) are highly temperature sensitive; the cytosolic enzyme was purified without the use of a previously reported method for Rhodospirillum rubrum family I PPase that involved heating the crude extract at 60 °C for 5 min. The C. sphaeroides enzyme lost all activity under these conditions (Romero, García-Contreras & Celis, 2003; Celis et al., 2003), and the enzymes lost activity if no Co2+ was included during purification. Additionally, the C. sphaeroides enzyme shows activity at a wide range of pH values (8.5−9.5), and the catalytic parameters for PPi hydrolysis with Mn2+ or Mg2+ are the same (Celis et al., 2003). In the R. rubrum enzyme, free Mg2+ had an activating effect, while C. sphaeroides had a strong inhibitory effect (Celis et al., 2003). The activation energies of the family I R. rubrum enzyme and family II C. spaheroides enzyme are different; that of the R. rubrum enzyme is 67.3 kJ/mol, while that of the C. sphaeroides enzyme is 36.94 kJ/mol (Ordaz et al., 1992; Celis et al., 2003). These results are in agreement with the temperature sensitivity of the C. sphaeroides enzyme. Interestingly, the c-PPase activity in C. sphaeroides and Rhodobacter capsulatus is not inhibited significantly by NaF, imidodiphosphate, or methylene diphosphate, which are strong inhibitors of family I enzymes (Celis et al., 2003). Nevertheless, Methanococcus janaschii has been demonstrated to protect against NaF inhibition (Kuhn et al., 2000). There is limited evidence regarding the inhibition of family II enzymes by methylene diphosphate or imidodiphosphate in addition to the B. subtilis enzyme.

In a follow-up study, Sarmina, Peña Segura & Celis (2017) demonstrated that c-PPases from photosynthetic bacteria can hydrolyze PPi in the absence of metal cations. In experiments with purified enzymes that were desalted by dialysis and Sephadex G-25, the family I enzyme lost all activity if no Mg2+ was present, but family II enzymes retained the same approximate activity (Sarmina, Peña Segura & Celis, 2017). Additionally, Sarmina, Peña Segura & Celis (2017) demonstrated that the metal bound to the enzyme is essential for activity since EDTA completely inhibited the activity at 100 µM EDTA. The presence of the chelator does not affect the dimeric nature of the C. sphaeroides enzyme, as demonstrated by native PAGE analysis (Sarmina, Peña Segura & Celis, 2017). The depletion of the metal ion is not restored by incubating the enzyme with the Mg2+ ion; thus, the strong dependence of Co2+ on enzyme activity is unremarkable, and Co2+ seems to be the key to EDTA resistance in this enzyme. The crystal structure of the C. sphaeroides enzyme sheds light on the involvement of Co2+ in the catalytic activity of this enzyme and its role in catalysis in the absence of Mg2+. The metal ratio per subunit is relevant in the catalytic steps of enzymes (Zyryanov et al., 2004).

In family II c-PPases from photosynthetic bacteria, once the Co2+ ion is chelated, the activity cannot be restored even though the oligomeric state is maintained (Sarmina, Peña Segura & Celis, 2017), which is consistent with the finding that in the B. subtilis enzyme, the active site has three metal ions that coordinate the PPi oxygens and results in high mobility in the active sites of family II enzymes (Fabrichniy et al., 2007). The metal ratio has been determined by proton beams to induce X-ray emission (PIXE), which allows one to know the exact number of metal ions per subunit. Solís et al. (2011) demonstrated that in family II c-PPase from Rhodobacter capsulatus, the enzyme possesses three Co2+ ions per enzyme, which contrasts with the finding in PDB 1K23, where the B. subilis enzyme contains only two Mn2+ ions and has been proposed to form an activated water molecule that bridges the two ions and is involved in nucleophilic attack on the substrate (Ahn et al., 2001).

Further research is needed to elucidate the different mechanisms involving the metal ions bound to family II c-PPases. A tool for determining the available Mn2+ and Co2+ in family II enzymes is arsenazo(III) (2,2′-[1,8-dihydroxy-3,6 disulfo-2,7-naphthalene-bis(azo)]dibenzenearsonic acid) reactions (Zyryanov & Baykov, 2002). This technique allows the quantification of the amount of free metal ions tested as well as those derived from hydrolyzed enzymes, which facilitates quantification and ratio estimation.

When the first family II enzyme was discovered (Young et al., 1998; Shintani et al., 1998), it took almost ten years to discover novel family II enzymes (Jämsen et al., 2007). The novel enzymes have extended domains that are regulated by ADP and AMP between the N- and C-terminal domains that form the mobile active site described in previous studies (Kajander, Kellosalo & Goldman, 2013). These enzymes contain a cystathione β-synthase (CBS) domain and are regulated by AMP and ATP; removing the CBS domain favors its activity and eliminates its regulation by phosphorylated nucleotides (Salminen et al., 2014).

Using bioinformatic analysis, as shown in Fig. 7A shows the alignment of examples from family II enzymes, including enzymes from photosynthetic bacteria, Asgard archaea and other archaeal organisms, and, surprisingly, from the intestinal parasite G. intestinalis. The family II CBS domain-containing enzymes were thought to belong exclusively to prokaryotes but are now reported in other protists, such as Emiliania huxleyi, Fragilariopsis cylindrus, Giardia lamblia (UniProt ID V6TM56 and V6U0W2), Guillardia theta, Salpingoeca rosetta, Symbiodinium minutum, and Thalassiosira pseudonana, and four green algae, Bathycoccus prasinos, Micromonas commoda, Ostreococcus lucimarinus, and Ostreococcus tauri (Baykov et al., 2017). As Baykov et al. (2017) reported, the ever-growing databases may contain enzymes with novel features that deserve further study. We analyzed the structure of these enzymes with a two-domain active site containing extensions with CBS domains. In Fig. 7B, we show the models retrieved from the AlphaFold2 database using a rainbow color scheme. The figure indicates that the N- and C-terminal ends form the active site and are between the regulatory domains. We could not find a report describing the presence of a family II c-PPase and an m-PPase. However, in the case of Thermotoga maritima, the presence of a family II enzyme (UniProt ID Q9WZ56) also correlates with the presence of a K+-stimulated m-PPase (UniProt ID Q9S5X0).

The findings for T. maritima led us to search for pathogenic organisms that may also encode a family II protein and a single m-PPase. Our survey revealed that Clostridium tetani encoded a family II c-PPase (UniProt accession number A0A4Q0VED8_CLOTA) and a putative K+-stimulated m-PPase (UniProt accession number A0A4Q0V5P6_CLOTA). In Fig. S1, we show that the structure of the cytoplasmic family II PPase contains the regulatory CBS domain and present a structural alignment showing that the catalytic domain is highly similar to that of the B. subtilis c-PPase (Fig. S1B). Additionally, m-PPase displayed a fold consistent with that of H+ m-PPases. The role of m-PPase in C. tetani physiology remains to be determined. Nevertheless, the enzyme contains only three intrinsic tryptophan residues. The highly conserved Trp602 plays a role in stabilizing the protein structure (Chen et al., 2014). These enzymes and their role in metabolic dynamics in this organism remain to be elucidated; specifically, c-PPase is highly active (Chen et al., 2014). These enzymes may be targeted for inhibition.

Table 1 Identification of family II PPases having cystathionine β-synthase domains.

Description	Scientific name	Max score	Total score	Query cover	E value	Per. Ident	Acc. Len	Accession	
Putative Manganese-dependent inorganic pyrophosphatase [Giardia intestinalis]	Giardia intestinalis	436	436	89%	3.00E−147	100.00%	670	XP_001709328.2	
Putative manganese-dependent inorganic diphosphatase [Deltaproteobacteria bacterium]	Deltaproteobacteria bacterium	115	115	64%	3.00E−25	37.50%	549	RME41472.1	
Putative manganese-dependent inorganic diphosphatase [Geothermobacter hydrogeniphilus]	Geothermobacter hydrogeniphilus	101	101	64%	3.00E−20	36.42%	555	WP_103114773.1	
Putative manganese-dependent inorganic diphosphatase [Candidatus Dependentiae bacterium]	Candidatus Dependentiae bacterium	89	89	63%	4.00E−16	33.56%	550	MBP7651980.1	
Pyrophosphatase [Verrucomicrobiota bacterium]	Verrucomicrobiota bacterium	88.6	88.6	63%	6.00E−16	30.00%	550	MCD6048652.1	

The family II CBS-containing enzymes shown in Fig. 7 suggest important differences among the enzymes, specifically in the G. intestinalis enzyme, where two loops are modeled. Complementary to the overall structure comparison, in Table 1, we show the data from BLAST analysis using the G. intestinalis sequence, and the resulting hits indicate that the CBS domain is variable among proteins. Conducting the same analysis with HMMR (Fig. 8A) revealed that the domain is present mostly in eukaryotic cells but has low conservation significance. As shown in Fig. 8B, via bioinformatic analysis, structural comparison with the B. subtilis enzyme revealed that the catalytic residues are in the same pocket as those in the B. subtilis enzyme. Additionally, the overall comparison between B. subtilis c-PPase and the selected examples shown here (Fig. 8C) indicates that the catalytic domain is highly conserved, and except for two loops in the G. intestinalis enzyme, the CBS regulatory domains are structurally conserved. These two loops are either relevant for the enzyme’s structure or are intrinsically disordered regions.

Figure 8 Giardia intestinalis enzyme contains a longer N-terminal domain and an internal loop that is absent in most CBS-domain-containing family II enzymes.

(A) compares the G. intestinalis enzyme and B. subtilis family II PPase. Here, G. intestinalis is shown in a rainbow color scheme, and comparisons between the G. intestinalis (in gray) and B. subtilis (in magenta) enzymes are shown (RMSD = 2.91, TM-score = 0.286). The locations of the DHH signature and SRKKQ C-terminal signature are shown in the same panel. In (B), the distribution of hits using HMMER is concentrated in eukaryotic organisms and a few examples of bacterial organisms. In (C), the AlphaFold2 dimer model of the G. intestinalis enzyme is shown, with each monomer indicated in blue and red, which is inconsistent with the experimental evidence suggesting that the catalytic domain is the center for dimers that form tetramers. In (D), a comparison of the selected examples is shown. The left image shows the overall alignment in gray, except that the B. subtilis enzyme is shown in magenta. The right image shows the alignment of Giardia intestinalis, Deltaproteobacteria bacterium, Geothermobacter hydrogeniphilus, Candidatus Dependentiae bacterium, Verrucomicrobiota bacterium, and B. subtilis.

Family II enzymes have been shown to form dimers, which are important for catalytic reactions, as shown for B. subtilis enzymes (Ahn et al., 2001; Shizawa et al., 2001). A search in the PDB provided no results for the crystal structures of these enzymes. Thus, the exact assembly mode and symmetry at the crystal structure level remain to be elucidated. However, Dadinova et al. (2020) reported the formation of tetrameric structures by small-angle X-ray scattering of purified recombinant proteins. Using the G. intestinalis sequence, the dimeric model is inconsistent with the findings in bacterial c-PPases determined by Dadinova et al. (2020), who reported that the interaction interface is the catalytic domain, not the regulatory domain. In the model, the interface is the regulatory domain for G. intestinalis (Fig. 8D). Long loops are not present in other bacterial CBS family II c-PPases and interfere with the prediction, suggesting that experimental determination of the crystal structure is needed. Anashkin et al.’s (2020) model of the Clostridium perfringens c-PPase showed that the tetrameric structure is a dimer similar to that of G. intestinalis (Fig. 8D). The catalytic domains, in turn, form the tetrameric form of the enzyme in its active form (Anashkin et al., 2020).

Additionally, Zamakhov et al. (2023) confirmed the tetrameric nature of these enzymes by purifying and characterizing the enzyme of Desulfobacterium hafniense by transmission electron microscopy and biochemical approaches. They found that a highly flexible region is responsible for limiting the crystallization of these enzymes, which is consistent with the findings of G. intestinalis. More studies are needed to assess the role of the flexible region and the regulatory mechanism of CBS-containing enzymes.

Thus, the future of research into enzymes bearing CBS domains is to obtain good-quality crystal structures and identify the dimerization surface in these proteins. Additionally, the DHH and RKKQ signatures are present in this enzyme. This local folding is consistent with the catalytic activity of a family II enzyme (Fig. 8B) (Shizawa et al., 2001). Future mutagenesis studies can uncover the role of the SRRKQ signature in CBS domain-containing c-PPases.

The finding of homologous CBS-containing c-PPases in the TM6 candidate phylum and Verrucomicrobia bacterium, two potential groups linking bacteria and higher eukaryotic organisms (Devos, 2021), suggests that the evolution of highly regulated CBS domain-containing proteins may ultimately involve bacterial CBS c-PPases as ancestors and thus a novel linage of proteins that are regulated by internal inhibition, since the CBS domain is present in enzymes and transporters in all kingdoms of life (Baykov, Tuominen & Lahti, 2011) and is linked to important conformational changes in the enzyme, resulting in strong regulatory flexibility (Ereño Orbea, Oyenarte & Martínez-Cruz, 2013). Thus, the presence of these genes in this group may shed light on the regulatory and metabolic mechanisms needed for these organisms to thrive.

Identifying CBS-containing c-PPases by searching for novel enzymes in sequenced genomes (from cultured organisms to environmental samples) is a much-needed task. The discovery of novel enzymatic systems has revealed the origin of metabolic pathways. However, detailed biochemical analysis is the only approach for assessing certain active pathways. PPi- and CBS-containing enzymes play a role in controlling glycolytic enzymes (for example, phosphoenolpyruvate carboxyltransferase); it has been shown that the binding of regulatory nucleotides and polyphosphates results in the control of metabolic enzymes under low-energy conditions and thus reduces or inhibits biosynthetic reactions, counteracting the effect of PPi on the progression of synthetic reactions (Müller et al., 2012; Baykov et al., 2017).

The core metabolic toolbox of eukaryotic cells comprises 50 enzymes (Müller et al., 2012), mostly dependent on phosphorylation reactions. Müller et al. (2012) proposed that this basic core is reduced in parasitic organisms. This reduction is also found in microorganisms that thrive in sugar-rich environments, such as parasitic protists. Organelles associated with energy generation have also been minimalized into mitosomes or hydrogenosomes, and the close link between polyphosphates and other phosphorylated molecules acts as an alarm signal (Reeves, 1984; Müller et al., 2012; Michels et al., 2006; Wimmer, Kleinermanns & Martin, 2021). Additionally, Danchin, Dondon & Daniel (1984) reported that 2-ketobutyrate acts as an alarmon in E. coli, which clearly points to specialization in a specific environment and may be linked to the origin of mitochondria and metabolic pathways in higher eukaryotes. The ability of G. intestinalis to encode a family II c-PPase may be related to cytosolic ATP synthesis and is the first step toward the complex regulatory mechanism in which the accumulation of PPi and polyphosphates acts as a ratchet pawl to prevent the halting of synthetic pathways (Lloyd, Ralphs & Harris, 2002).

Using the tool Fold Seek (van Kempen et al., 2023), we searched for structures such as PDB 1WPM, corresponding to the dimeric form of the B. subtilis family II c-PPase, and similar structures were also found in Enterococcus faecium (UniProt A0A133CPF0) and S. aureus (UniProt Q2FWY1), which possess a typical Mn-dependent family II c-PPase. However, using the same tool, we found that Trichuris trichiura (whipworm) (UniProt A0A077ZGU3), Plasmodium falciparum (UniProt Q8IM69), and Trypanosoma cruzi (UniProt Q4DJ30), among other hits, may possess an Mn-dependent family II PPase. Similarly, other exopolyphosphatases may exhibit the same folding and perhaps the same biochemical activity in higher organisms, suggesting that c-PPases may be the intermediate evolutionary ancestors of other phosphate metabolism enzymes in higher eukaryotes. The same analysis revealed that 37 PDB entries had similar folds and, in all instances, were structures of family II c-PPases. One example is PDB 4PY9, the crystal structure of an exopolyphosphatase-related protein from Bacteroides fragilis, which has a conserved fold and sequence with the B. subtilis family II c-PPase. Overall, proteins that may be related to family II c-PPases still need to be identified. Cumulative evidence indicates that these enzymes are not restricted to bacteria and are found in microbial eukaryotes, as previously suggested (Baykov et al., 2017).

However, the identification of c-PPases is still lacking. In the inorganic c-PPase of Medicago trunculata (MtPPA1), an N-terminal sequence of 29 to 32 residues (MSEETKDNQRLQRPAPRLNERIL) was missing in the crystal structure, and it was hypothesized that this sequence targeted the enzyme to the mitochondria due to its highly similar sequence and properties of a signal peptide (Grzechowiak et al., 2019). The N-terminal fragment is reported to be sequentially removed since the sequence missing in the crystal is variable and seems to be dependent on the actual activity of the enzyme (Grzechowiak et al., 2019). BLASTp analysis of the N-terminal fragment from MtPPA1 indicated that some plant processing and moonlighting functions of cytoplasmic PPases may play a deeper role in plant biology, as PPi plays a central role in sugar accumulation.

Membrane-Bound PPases

H+ or Na+ pump m-PPases are enzymes linked to transmembrane electrochemical potential of an ion gradient that provide inexpensive cell fuel. However, we think that the topic of m-PPases has a deeper meaning in the biology of the organisms that encode these enzymes, specifically in terms of the Na+/K+ ratio and the acidification of subcellular structures such as in the case of protists, the acidocalcisome, and the vacuole in plants. In this regard, Dibrova et al. (2015) provided a strong case for the relevance of Na+ and K+ homeostasis and a link to regulating sodium transport. In their seminal work, Dibrova et al. (2015) summarize the role of the prevalence of K+ ions in the cytoplasm over that of Na+ ions and argue that most organisms have a [K+]/[Na+]>1. This property is linked to metabolically active cells, and the ratio is decreased and even reversed in stationary-phase microorganisms (Fagerbakke, Norland & Heldal, 1999; Dibrova et al., 2015). Perhaps the abundance of organisms that strongly depend on K+ and Na+ imbalances is related to the ancient origin of cells that thrived in a K+-rich environment. This idea is reinforced by the numerous cell processes that are dependent on the K+ and Na+ concentrations, where the K+ concentration is crucial for the activity of key enzymes such as the translation factor EF-Tu, the recombinase RadA, the chaperonin GoroEL/Hsp60, diol dehydratases, and S-adenosyl methionine synthase, among others, and the presence of specific protein domains that require K+ ions to be active (Dibrova et al., 2015).

The above line of thought is strongly linked to the current knowledge of bacterial m-PPases that either pump H+ or Na+ to the periplasmic space (and in some cases both), and the activity is either K+-activated or K+-insensitive (Nordbo et al., 2016). The work of Nordbo et al. (2016)] clearly shows the existence of two subtypes of enzymes capable of transporting Na+ and H+ ions in response to Na+ ions in the environment and suggests that structural differences are relevant for assessing the role of m-PPases in both bioenergetics and transport activities (Nordbo et al., 2016), indicating convergence from Na+-pumping enzymes to the two types of unique H+ or Na+ pumps or cotransporters (H+/Na+). In Fig. 9, a sample of m-PPases from photosynthetic bacteria, archaea, and plants is in good agreement with the report by Nordbo et al. (2016), in the sense that the groups in bacteria are split into two subgroups and, in the case of archaea enzymes, are in the middle for well-characterized enzymes from both archaea and plants. These examples require experimental analysis for the transport of H+ or Na+. Plant enzymes are less divergent than their bacterial counterparts, and in this analysis, the results agree with the specificity of their function in vacuole and cytosolic pH homeostasis since there is coordination between the membrane PPase and other transporters needed for pH homeostasis (Cosse & Seidel, 2021). Additionally, as shown in the bioinformatic analysis in Fig. 9A shows that the enzymes have highly conserved overall physicochemical properties, and the distribution of conserved critical residues agrees with the properties of these enzymes. Additionally, as shown in Fig. 9B, the enzymes have a high degree of conservation. They are clustered in UMAP (McInnes et al., 2018), suggesting that in the sampled examples shown here, all enzymes are similar in sequence, but perhaps small variations in the N- or C-terminal ends may play additional regulatory roles.

Figure 9 Examples used for reconstructing the evolutionary distribution of m-PPases.

Phylogenetic tree for evolutionary analysis via the maximum likelihood method. The evolutionary history was inferred by using the maximum likelihood method and the JTT matrix-based model (Jones, Taylor & Thornton, 1992). The tree with the highest log likelihood (−17,479.90) is shown. The percentage of trees in which the associated taxa clustered together is shown below the branches. Initial tree(s) for the heuristic search were obtained by applying the neighbor-joining method to a matrix of pairwise distances estimated using the JTT model. The tree is drawn to scale, with branch lengths measured as the number of substitutions per site. This analysis involved 18 amino acid sequences. There were 957 positions in the final dataset. Evolutionary analyses were conducted in MEGA11 (Tamura, Stecher & Kumar, 2021). Manually, plant m-PPases are indicated with a green background. The numbers indicate the groups selected to show the predicted structures in Fig. 11.

The role of the transmembrane helices and the catalytic domain has been explored previously (Kellosalo et al., 2012). Transmembrane helices 5 and 6 close the active site and form a transient state that leads to hydrolysis, and transmembrane helices 11 and 12 move down to open the exit channel; then, Na+ (in this case) escapes to the extracellular medium while the active site is closed until hydrolysis is completed (Kellosalo et al., 2012 and for review, Kajander, Kellosalo & Goldman, 2013). The catalytic mechanism of these enzymes has recently been characterized as asymmetric catalysis (Anashkin et al., 2021), where the enzyme combines direct coupling of the substrate with conformational coupling to pump ions across the membrane. The fate of ion pumps in m-PPases and the differences in their mechanisms are still emerging as enzyme mechanisms that need further research.

One such example is the m-PPase of Rhodospirillum rubrum, a purple nonsulfur bacterium. In this organism, m-PPase is capable of PPi hydrolysis but also uses the proton gradient derived from photosynthetic electron transport to synthesize PPi (Baltscheffsky et al., 1966 and Baltscheffsky, 1967). In the case of R. rubrum, m-PPase has been extensively characterized, and the use of different methods to determine the stoichiometry of the relationship between PPi hydrolysis and the ratio of H+ translocated has been determined to be 2 H+ per PPi hydrolyzed (Sosa & Celis, 1995), which implies a relationship between ATP synthesis and the divergence of H+ m-PPases. In R. rubrum, the synthesis of PPi occurs at lower ΔpH values than in the vacuolar systems (Sosa & Celis, 1995), suggesting different kinetic activities linked to the enzyme itself and the environment in the chromatophores of R. rubrum. Additionally, this enzyme can hydrolyze either Zn-PPi or Mg-PPi, and free Zn is needed to activate the enzyme within a range of 7.5 pM. This enzyme is inhibited not by NaF but by 1-butanol and methylene diphosphate (Romero & Celis, 1995). Finally, this enzyme was studied to determine the effect of lipid charges. These findings suggest that none of the kinetic parameters are modified by either positively or negatively charged lipids in the chromatophores (Sosa & Celis, 1993). The study of the effect of the membrane environment on m-PPases is an area for future research; we propose that assessing the impact of lipid-disturbing drugs on the activity of m-PPases may be important for controlling pathogenic organisms encoding this type of enzyme.

The high similarity of membrane-bound enzymes between ancestral and modern enzymes also results in structural conservation. Figure 10 shows the models of the subgroups identified by bioinformatic analysis, and Fig. 11 shows the high conservation of the 15 to 16 transmembrane helices (Baltscheffsky, Schultz & Baltscheffsky, 1999). As shown in Fig. S3, the results of the bioinformatic analysis of topological diversity in the transmembrane distribution of the sampled m-PPases suggested that the diversity was localized in the loop regions of these enzymes. Figure 11 shows that the first set of enzymes contains an extension in the C-terminal end that may have a regulatory role. This may be related to the lipid composition of the membrane on each cell expressing this type of enzyme. The architecture of these enzymes in the cell membrane (or organelle membrane) is a relevant field to explore.

Complementary to this review, we browsed for homologous proteins with longer sequences in UniProt and attempted to determine whether we could find a membrane-bound enzyme with odd specificity. The first criterion for the search was to look for proteins that were significantly larger. Using UniProt, with the keyword ‘membrane pyrophosphatase’ and a filter for length, we identified a predicted K+-insensitive m-PPase from Paucibater sp. KCTC42545 of 810 amino acids (UniProt A0A0U2VTE8), which is larger than the previously identified membrane-bound PPases. Other examples of the same genera were found with a long C-terminal extension but not those identified from the KCTC42545 strain. In Fig. S4A, the AlphaFold2 model shows a 110-residue domain at the C-terminal end. As shown in Fig. S4B, HMMER analysis identified this region as an OmpA-like domain. In the same panel, the domain is not continuously homologous but shows an interesting similarity to the extracellular domain of OmpA. Additionally, it is a domain exclusively present in bacterial proteins, as shown in the taxonomic distribution in HMMER. In Figs. S4C, and S4D, the comparison between the two proteins shows limited homology to the three β-strands and part of a helix. Table 2 shows a sample of BLAST hits of homologous proteins that either had an extra domain in the C-terminus and/or were longer than the usual m-PPases. The role of this additional domain in the activity of m-PPases remains to be elucidated.

The transcriptional regulation of m-PPases has been studied in a limited number of examples and has mostly focused on plants since the expression and overexpression of m-PPases have been shown to increase drought resistance in plants (see the following section). In bacteria, little is known about the regulatory mechanisms controlling the expression of both soluble and m-PPases. Interestingly, in R. rubrum, under high-energy conditions, m-PPase is not expressed, while c-PPase is expressed at high levels (López-Marqués et al., 2004). Under low energy levels (fermentative growth conditions) and high salt stress (1 M NaCl), m-PPase becomes highly expressed (López-Marqués et al., 2004). This is also consistent with findings in algae and plants, where m-PPases are also induced under stress conditions (Carystinos et al., 1995; Fukuda et al., 2004; Serrano et al., 2007). The expression pattern is dependent on RegA, a key transcriptional regulator that belongs to the two-component system RegA–RegB and participates in the regulation of genes involved in photosynthesis and inorganic carbon and nitrogen assimilation (Joshi & Tabita, 1996; Qian & Tabita, 1996). The finding that RegA also controls the expression of photosynthetic pigment genes and m-PPase suggests the existence of primordial photosynthesis-based bioenergetics in ancestral anaerobic prokaryotes that influence the central role of m-PPase in the bioenergetics of these organisms (Serrano et al., 2007). This hypothesis was confirmed by García-Contreras, Celis & Romero (2004), who showed that under low-energy growth conditions, a mutant of m-PPase is unable to sustain growth, thus elegantly demonstrating that although PPi synthesis is, in most cases, the driving force for synthetic reactions, in the case of R. rubrum, it is needed to fuel biosynthetic reactions as an energy source. Importantly, it has been proposed that family I c-PPases, which are less active, are commonly found in organisms that have an m-PPase. However, we showed that there are exceptions where the more active family II enzymes are found in organisms encoding m-PPases, thus complicating the landscape of the function and regulation of m-PPases.

Figure 10 A sequence comparison of the analyzed m-Pases is shown in Fig. 9.

(A) shows the alignment viewer analysis, the upper figure in the MSA color scheme, and the charged and hydrophobic residues in the lower figure, as in previous figures. The enzymes are highly conserved, as evidenced by the 2D pairwise alignment (B), and the UMAP space distribution is more homogeneous than that for the family I and II cytoplasmic enzymes.

The co-occurrence of family I and m-PPases has been reported extensively. However, to our knowledge, the presence of m-PPases and family II cytoplasmic enzymes has not been explicitly reported. After exploring UniProt entries, we found that the co-occurrence of family II enzymes and m-PPases is not rare. For example, Clostridium tetani encodes an m-PPase (UniProt entry Q898Q9) and a cytoplasmic family II PPase (UniProt entry Q894A3). Additionally, in Thermotoga maritima, a family II protein (UniProt entry QWZ56) is found along with an m-PPase (UniProt entry Q9S5X0, PDB ID 4AV3), and in this case, the m-PPase has been studied in detail and its structure has been solved (Belgorod et al., 2005; Kellosalo et al., 2012; Li et al., 2016). In the case of archaea, the following question remains to be answered: Are there m-PPases with either family I or II cytoplasmic PPases that contribute to understanding the coexistence and evolution of the two PPases?

The presence of multiple PPases poses an interesting question: Is the presence of more than one cytoplasmic or m-PPase redundant? Or is it related to environmental cues that are solved by different mechanisms? One example of such a condition is the ciliate protist Philasterides dicentrarchi. This organism encodes two m-PPases: one seems to be stimulated by K+, and the other appears insensitive to K+ (Folgueira et al., 2021). The enzymes contain 746 and 810 amino acids (AAs) and show 33.5% identity and 51.2% similarity, with different numbers of transmembrane helices, which is consistent with the findings for vacuolar m-PPases.

Interestingly, the expression patterns of the two isoenzymes in P. dicentrarchi are highly similar, but they are more highly expressed during the endoparasitic phase and are repressed in the presence of antiparasitic drugs (Folgueira et al., 2021). The role of these two enzymes in this parasite remains to be elucidated. Nevertheless, the authors suggest that they may participate in invasion and, perhaps, the parasitic stage of this opportunistic fish parasite.

Similarly, the vacuolar m-PPase of Toxoplasma gondii participates in cathepsin L processing; thus, the null mutant exhibited a less virulent phenotype in mice (Liu et al., 2014). The proposed mechanism is that osmoregulation, but not Ca2+ homeostasis, is the key factor for extracellular survival. Thus, vacuolar m-PPase in T. gondii is related to osmoregulation during invasion of the host (Liu et al., 2014).

Computer multiscale simulations have recently addressed the complexity of these enzymes, and the conserved structure found in m-PPases (the number of transmembrane helices and the number of active site structural transitions) indicates that the lipid microenvironment is a determinant of both catalysis and protein dimeric structure (Holmes, Goldman & Kalli, 2022). Further research is needed to correlate the membrane composition of organisms encoding m-PPases, the activity of these enzymes, and their biological role. Holmes, Goldman & Kalli (2022) demonstrated that the lipid environment contributes to intersubunit communication and the current catalytic asymmetry model. For the three proteins studied (from T. maritima, V. radiata, and Clostridium leptum), the enzymes preferentially interact with anionic lipids rather than with neutral lipids (Holmes, Goldman & Kalli, 2022), which is important given the nature of the organisms that have been found to encode these enzymes, but the environmental conditions under which these organisms thrive are different. The work by Holmes, Goldman & Kalli (2022) proposes that experimental work is needed to demonstrate the specific interactions with lipids and the residues (that they report are not conserved) present in the protein.

Finally, studying the inhibitory mechanisms and identifying novel molecules that target m-PPases is relevant for future antimicrobial therapeutics for bacteria and protist organisms. For example, the identification of inhibitors such as those derived from isoxazole fragments is relevant because they are nonphosphorus compound inhibitors (Johansson et al., 2020). Johansson et al. (2020) demonstrated that two isoxazole compounds reduced the activity of m-PPase and Plasmodium falciparum proliferation at an IC50 = 55 µM. Using diverse compounds to inhibit m-PPase in parasitic organisms has expanded the arsenal to prevent resistance and ensure correct treatment.

As stated above, R. rubrum m-PPase plays a central role in the bioenergetics of this photosynthetic bacterium. Some inhibitors, such as DCCD, have been identified as blocking agents of H+ translocation (Baltscheffsky, 1968). Celis, Escobedo & Romero (1998) found that triphenyltin, a compound that has also been shown to block H+ translocation, has been identified as an inhibitor of the m-PPase of R. rubrum. Triphenyltin was also shown to be inhibited at lower concentrations than DCCD and achieved total inhibition. Additionally, Celis, Escobedo & Romero (1998) reported that this inhibition is partially prevented by 2-mercaptoethanol and fully prevented by dithioerythritol (DTE). Overall, the inhibition studies of m-PPase are relevant for the future design of compounds capable of reducing or inhibiting m-PPase in pathogenic organisms, and extensive studies are needed to assess the role of m-PPase in the bioenergetics of these compounds, considering that the effect may be hampered by reducing agents or other cellular components. Tripenyltin is highly toxic, but derived compounds with reduced toxicity could be used as inhibitors.

The study of the evolution of m-PPases has struggled to pinpoint them either as an inexpensive energy source or as a precursor to more complex enzymes. A report by Baltscheffsky & Persson (2014) on the finding of an early version of the m-PPase suggested that in the archaeon Candidatus Korarchaeum cryptofilum, the m-PPase contains features consistent with a H+ pump. The authors suggest that the earlier life forms may have an acidic pocket in the catalytic domain and, thus, it is compatible with a pump of H+ rather than Na2+ ions. Archaea, such as Asgard archaea, have been proposed to be early divergent organisms in terms of eukaryotic diversification (Devos, 2021). In Candidatus Lokiarchaeum, the encoded m-PPase is an H+ pump with features consistent with an enzyme insensitive to K+ (UniProt A0A0F8VHH2) and contains genes encoding family I (UniProt A0A0F8W2R4) and family II (UniProt A0A0F8XQ12) c-PPases, which is important to discuss here since the Asgard archaea are intermediates in the origin of eukaryotic organisms. The observation of the three types of enzymes suggested that the origin and evolution of m-PPases are more complex than previously thought, especially as they spread through more eukaryotic organisms.

Overall, although m-PPases are simple and their overall structure and enzymatic mechanisms are well known, there are still many examples of m-PPases that deserve further research.

Are m-PPases present in fungi?

With the report of PPase activity bound to the membrane in S. carlsbergensis, we questioned whether other fungi may encode an m-PPase. It has been assumed that no m-PPase homologs exist in protists, yeast, fungi, or mammals. However, m-PPase is now recognized as a relevant enzyme in protists (Serrano et al., 2007), especially in choanoflagellates (Serrano, 2004). The enzyme was lost in the early evolution of animals and fungi, as the enzyme might have been lost in a choanoflagellate ancestor of higher eukaryotes (King & Carroll, 2001; Serrano et al., 2007).

Figure 11 The predicted structure for the examples of m-PPases is analyzed in Fig. 9.

The number indicates the corresponding cluster of branches shown in the phylogenetic reconstruction in Fig. 10. Structures are shown in a rainbow color scheme. The right corner shows the structure of the V. radiata enzyme.

Table 2 Identification of H+ PPases having a C-terminal OmpA-related domain.

Description	Scientific name	Max score	Total score	Query cover	E value	Per. ident	Acc. len	Accession	
Sodium-translocating pyrophosphatase [Paucibacter sp. KCTC 42545]	Paucibacter sp. KCTC 42545	215	215	100%	7.00E−63	100	810	WP_058719013.1	
Sodium-translocating pyrophosphatase [Burkholderiaceae bacterium]	Burkholderiaceae bacterium	210	210	99%	2.00E−62	99.08	571	MBY0237279.1	
Sodium-translocating pyrophosphatase [uncultured Paucibacter sp.]	uncultured Paucibacter sp.	205	205	100%	5.00E−59	94.55	810	WP_295085243.1	
Sodium-translocating pyrophosphatase [Pseudomonadota bacterium]	Pseudomonadota bacterium	148	148	91%	8.00E−39	77.23	643	MBS0440426.1	
Sodium-translocating pyrophosphatase [Betaproteobacteria bacterium]	Betaproteobacteria bacterium	141	141	90%	3.00E−36	74	818	MBK9684042.1	

An intriguing question regarding the evolutionary distribution of these enzymes is that they are not found in algae but in Arabidopsis, mung bean, and barley. Additionally, m-PPases are found in archaea, which may provide a link for the presence of these enzymes in higher eukaryotic organisms. Nevertheless, these enzymes have been found in all organisms except fungi and mammals, and further insight is needed to clarify the evolutionary link of these enzymes in eukaryotes.

However, as shown in Fig. 12A, we conducted a BLASTp search in Fungal DB (Basenko et al., 2018) using Vigna radiata (mung bean) membrane-bound PPase and identified 61 proteins with putative m-PPase sequences and structures homologous to those of V. radiata m-PPase; the last hits were either fragments of proteins or unrelated proteins. Additionally, the predicted structure of some examples from this analysis indeed presented a predicted structure with m-PPases (Fig. 12B and, for comparison, please see Fig. 2). The organisms encoding m-PPases include oomycetes and one example of an ascomycete, from which one example is a nonrelated enzyme. This suggests that with the increasing number of available genomes, the presence of m-PPases may open a new avenue for assessing the function of these enzymes in fungal organisms. For example, Pythium ultimum is a plant pathogen, so studying these enzymes may also shed light on the possible control of plant pathogens bearing this enzyme. Additionally, BLASTp was used to detect fragments with homology to membrane PPases. We collected the best and worst hits and searched for AlphaFold2 models (Fig. 12), and the results suggested that not only do the sequences show homology but also that the structural models are consistent with those of membrane PPases. Therefore, this enzyme is present in fungal genomes, but little is known about the expression pattern of this enzyme and its possible kinetic parameters in the fungal membrane. One remaining question is what role the m-PPase may play in fungi and why the enzyme was not lost in all these organisms (Holmes, Kalli & Goldman, 2019).

Figure 12 M-PPases are found in fungi.

(A) the table summarizes the results obtained from BLASTp analysis of the V. radiata m-PPase sequence via the BLAST server FungalDB. The red asterisk indicates the results searched in UniProt and retrieved from the Alphafold2 models; in all instances, the positive hits belong to the oomycote phylum except for Sordaria macrospora, which belongs to the Ascomycota phylum. The S. macrospora model was generated with the Alphafold2 collaborative environment (Mirdita et al., 2022). (B) the models of PPases corresponding to the indicated hits in (A) shows that the proteins not only exhibit sequence-level similarity with m-PPases but also have similar predicted folding. The lowest hits either are fragments with similarity to m-PPases, or the lowest hits have folding and sequence similarity to ABC transporters (both predicted structures are shown). The predicted fold change confirmed that the protein has a region similar to that of PPases, but the folding is unrelated to these enzymes. The latter two are close to PDB 6LR0, a bile salt exporter from humans, and PDB 6FN4, an ABDSB1 protein from humans.

One hypothesis is that the oomycete cell wall is different from that of true fungi because the former is composed of cellulose and β-glucans. In contrast, the latter contains both β-glucans and chitin as structural polysaccharides (Mélida et al., 2013; Gow, Latge & Munro, 2017). The donor sugar for the synthesis of both oomycete polysaccharides is UDP-glucose, which is synthesized by UDP glucose pyrophosphorylase, which acts on glucose-1-phosphate and UTP, generating UDP-glucose and PPi (Gow, Latge & Munro, 2017). Therefore, the active synthesis of the cell wall involves the constant hydrolysis of PPi generated during UDP-glucose synthesis. Since glucan and cellulose synthesis are performed by membrane-bound enzymes (Gow, Latge & Munro, 2017), it is feasible to hypothesize that membrane-bound PPases can prevent the accumulation of PPi, which remains to be proven experimentally. One putative m-PPase, the sole fungal species listed in Fig. 12, was identified in Sordaria macrospora. The search for putative orthologs in other fungi was unsuccessful, but they are highly similar to other enzymes from bacteria. The function of this apparent unique fungal m-PPase remains to be evaluated (Maeshima, 2000; Holmes, Kalli & Goldman, 2019). Additionally, this enzyme’s subcellular localization is key for determining its role in either cell wall biosynthesis or proton pumping activity in another cell compartment for an unknown process. These hypotheses remain to be explored experimentally.

PPases and Applications

As stated in the introduction, c-PPase is an essential enzyme needed to produce the major biomolecules in the cell. In this section, we present some applications of the PPases that have been studied that we consider relevant.

Molecular biology applications

In vitro transcription for generating large amounts of RNA is taking the central stage with the development of RNA-based vaccines and therapeutics (Perenkov et al., 2023; Malla et al., 2023) and the study of RNA structure −function relationships (Tersteeg et al., 2022). To produce large amounts of RNA, c-PPase plays a key role in preventing the accumulation of PPi as a byproduct and thus inhibiting the enzymes for transcribing the RNA, in particular by preventing the accumulation of Mg-PPi, since Mg2+ ions are essential cofactors for T7 RNA polymerase, reducing yield and thus preventing the precipitation of Mg-PPi (Tersteeg et al., 2022). Additionally, identifying and purifying novel enzymes for in vitro transcription has been a goal for increasing yield. An example of a versatile application is the enhancement of PCRs in the presence of thermostable enzymes such as the c-PPase from Thermococcus onnurineus NA1 (Li, Yang & Gao, 2022) and the increased activity of metabolic enzymes such as UDP-glucose and UDP-galactose (Li, Yang & Gao, 2022), which have important industrial applications such as DNA synthesis, DNA and RNA sequencing, and sugar-nucleotide synthesis. Thus, producing c-PPases in a cost-effective method that renders highly active enzymes that are easy to express and purify is desirable because of increasing interest in mRNA-based therapeutics.

Another highly relevant example for this review is the use of c-PPase to develop novel reporter systems. For example, Biswas et al. (2013) developed a screening method for the bacterial DNA primase DnaG, which is needed for chromosomal replication and is structurally distinct from its eukaryotic counterpart. DnaG is an RNA polymerase that releases PPi. By coupling PPi hydrolysis, Pang, Garneau-Tsodikova & Tsodikov (2017) were able to use Mycobacterium tuberculosis primase and c-PPase to screen for inhibitors of both enzymes, selecting nine positive compounds, three of which were confirmed to be inhibitors of DnaG (Biswas et al., 2013). The method uses two purified enzymes: the primase transcribes an RNA primer, which in turn releases PPi; the c-PPase activity is detected by transforming malachite green from a yellow solution into a green solution. This is the first nonradioactive primase activity assay that has served to screen for inhibitory compounds for either enzyme. Thus, this is a tool for the search for novel antimicrobial agents for relevant pathogenic bacteria, a latent threat due to the high rate of antibiotic resistance.

Quantification of the effect of PPi on human health

With the above examples, the correct quantification of PPi has become important in biotechnology and health. For example, we recommend the review by Orriss (2020) on the role of extracellular PPi as a water softener. One example of the importance of PPi is its role in mineralization. Overall, there are several conditions in which the amount of PPi is relevant, especially in tissue calcification disorders (Ralph et al., 2022). A reduced concentration of circulating PPi, which is the main inhibitor of calcium hydroxyapatite deposition in soft connective tissue, leads to ankylosis (Ralph et al., 2022). Thus, improving PPi quantification in serum is a cornerstone for future diagnostic methods.

Measuring inorganic phosphate has been the standard method for quantifying PPase activity (Sumner, 1944). Thus, most methods rely on purified c-PPase to convert PPi into Pi.

A recent paper described the use of coupled enzymatic reactions using ATP sulfurylase to generate ATP and then quantified it by firefly luciferase activity (quantifying the bioluminescence) and included an internal ATP standard to correct for sample-specific interference since the authors aimed to determine the plasma levels of PPi (Lundkvist et al., 2023). This assay shows excellent precision and accuracy, and interestingly, the anticoagulant EDTA blocked the conversion of ATP into PPi in plasma after blood collection.

However, few studies have focused on improving the direct measurement of PPi. In this review, we propose that the use of nanotechnology in the future may further enhance the direct measurements of PPi. For instance, Zhang et al. (2023a; 2023b) developed a carbon quantum dot system in which highly versatile carbon quantum dots are coupled to nanotubes synthesized using metal–organic frameworks. The detection method worked via a “turn-off” mechanism, where Cu2+ ions quench the fluorescence from the quantum dots, and the detection is based on the decomposition of the metal–organic framework containing the Cu2+ ions, which are released by the coordination of the Cu2+ ions with PPi, leading to fluorescence recovery. The fold change in fluorescence determines the presence of c-PPase (human serum PPase) due to the hydrolysis of PPi (Zhang et al., 2023a). This method was proven to work not only for measuring PPase activity but also for measuring the amount of PPi in the sample. This is an alternative, more sensitive method for detecting PPi and PPase activity because of the low toxicity of the sensor components. A similar reporter system was described by Zhou et al. (2016) using bimetallic nanoclusters, where Cu2+ ions quench the high fluorescence emitted by an Au–Ag nanocluster, and the recovery of the signal is achieved by the coordination of PPi with the Cu2+ ion, which in turn allows the nanocluster to emit fluorescence. In the presence of PPase activity, the fluorescence is quenched by the hydrolysis of PPi, releasing the Cu2+ ion.

Overall, nanotechnology-derived sensing methods are powerful tools for the sensitive and accurate measurement of PPi and PPase activity and constitute a platform for the development of high-throughput screening methods for novel inhibitors of PPase enzymes for the treatment of a myriad of diseases (Tian et al., 2019).

PPases as biological targets

The search for novel PPases, specifically in pathogenic bacteria, protists, and fungi, is relevant for identifying novel treatment targets. One example is the identification, molecular cloning, and functional characterization of a gene encoding an H+-pyrophosphatase in the protozoan scuticociliate parasite Philasterides dicentrarchi, which infects turbot (Mallo et al., 2015). The enzyme resembles the plant vacuolar m-PPase and is highly sensitive to PPi analogs, inhibiting the growth of the ciliate parasite and resulting in a promising target for controlling scuticociliatosis.

The second area where PPi is important is human health. Numerous reports have assessed the role of the PPi or c-PPase in homeostasis. One example is the role of the PPi in tumor biology, which has been addressed extensively. Evidence suggests that it is an essential housekeeping enzyme that is overexpressed in tumors and is a key player in cell signaling, both in the regulation of its activation and in the role of tumor cell metabolism (Wang et al., 2022). The overexpression of c-PPase in tumors can potentially be targeted by anticancer drugs (Niu et al., 2021; Menteş & Yandım, 2023). One example of the role of c-PPase is the heightened expression of PPA1, the housekeeping isoform of c-PPase in humans, in colorectal tumors (Niu et al., 2023). Niu et al. (2023) reported that PPA1 is highly expressed in different cell lines and colorectal tumor samples and is concurrent with activation of the PI3K/Akt pathway. This report highlights the importance of activating the PI3K/Akt signaling pathway since this pathway controls cell proliferation and stemness properties in colorectal tumor cells.

Previously, Niu et al. (2021) obtained the crystal structure of human c-PPase (PDB 6C45) and determined that the active site has the possibility of binding JNK1-derived phosphor peptides, suggesting its possible role as a protein phosphatase, in agreement with previous observations of the interrelationships between JNK1 activity and PPA1 abundance (Wang et al., 2017). Structural models showing the binding of JNK1-derived peptides are a starting point for revealing novel functions of c-PPases. Additionally, the authors propose that PPA1 activation is a putative novel biomarker and a potential target to block tumor progression due to its important metabolic role (Niu et al., 2023).

PPase inhibitors

The discovery of novel inhibitors for PPase enzymes is not an easy task. The difficulty lies in designing inhibitors that selectively and effectively target the inorganic pyrophosphatase of a desired organism, for example, a bacterial or protist pathogen, without causing undesirable side effects or interfering with the activity of the host PPase or essential cellular processes and pathways. The complexity of the enzyme’s active site (and differences in each family of enzymes), its involvement in multiple cellular pathways, and the need for high specificity and low toxicity in off-targets pose significant challenges in inhibitor discovery and development. Overcoming these hurdles requires a deep understanding of the enzyme’s structure −function relationship with respect to the target organism and innovative approaches in drug design to achieve the desired therapeutic outcomes and, as stated previously, high-throughput methods.

Sodium fluoride is the main inhibitor of family I enzymes. However, the quest for novel inhibitors of family II enzymes is limited to the use of pyrophosphate analogs, such as imidodiphosphate, the most potent inhibitor, and the modest effect of other derived molecules (Zyryanov, Lahti & Baykov, 2005). Additionally, the analogs studied by Zyryanov, Lahti & Baykov (2005) inhibit family I enzymes. The structures of such analogs can be further engineered to find novel inhibitors. However, as stated in the previous sections, novel molecules may help uncover other inhibitors to target family II enzymes. In a paper published in 2017, the inhibitory effect of fructose-1-phosphate on the E. coli enzyme was described (Vorobyeva et al., 2017). The binding and inhibition maxima of fructose-1-phosphate were observed at 1.1 mM, and the authors suggested that fructose-1-phosphate exerts a regulatory effect on metabolism since a previous study showed the interaction of c-PPase with fructose-1,6-bisphosphate aldolase and 5-keto-4-deoxyuronate isomerase (Rodina et al., 2011). Thus, the analysis of protein −protein interactions in other organisms by pull-down experiments and the exploration of phosphate-sugar analogs may also provide fertile ground for finding novel inhibitors and identifying the specific role of c-PPases in cell metabolism, as shown here for E. coli. Another example of a novel PPase inhibitor that targets medically relevant microorganisms was reported by Pang et al. (2016), who described derivatives of 2,4-bis(aziridine-1-yl)-6-(1-phenylpyrrol-2-yl)-s-triazine, an allosteric inhibitor of bacterial PPases. The reported compounds inhibited the Mycobacterium tuberculosis enzyme by binding to an unprecedented target, the interface between monomers, in a species-selective manner. Further work will uncover novel compounds that may point to specific treatments for bacteria and perhaps other organisms in the future.

For membrane PPases, triphenylitin (Celis, Escobedo & Romero, 1998) was demonstrated to inhibit the activity of the m-PPase of R. rubrum. Other inhibitors have been found to increase the asymmetry of T. maritima activity (Vidilaseris et al., 2019). Vidilaseris et al. (2019) employed N-[(2-amino-6-benzothiazolyl) methyl]-1H-indole-2-carboxamide (ATC), which is a nonphosphate and nonsubstrate inhibitor of the m-PPase of T. maritima, and showed in structural studies that it inhibits an allosteric mechanism; thus, the asymmetry found in the dimers of m-PPase, i.e., the subunit alternating between substrate binding and catalysis, was revealed. Compounds similar to ATC exhibit reduced inhibition. With the cumulative information presented in this manuscript, we propose that the organisms encoding an m-PPase are relevant to the study of the effects of its inhibition on the physiology of pathogenic organisms, as shown by Vidilaseris et al. (2019); in the same study, no viability effect was found on Plasmodium spp. cells. Mechanistically, the structures of the other m-PPases differed in terms of the location of the ATC target site, which explains the difference in the effect on the T. maritima enzyme. Further research is needed to expand the applicability of m-PPase inhibition in pathogen control.

The role of PPases as biotechnological tools

The heterologous expression of m-PPases is a novel avenue for biotechnological applications. There are some limited examples of heterologously expressed m-PPases in microorganisms, resulting in a detectable phenotype. One example is the complementation of a distinct type of PPi-dependent proton pump that can efficiently substitute for the V-ATPase of S. cerevisiae in diverse physiological scenarios (Pérez-Castiñeira et al., 2011). The authors demonstrated that the overexpression of the K+-dependent proton pump m-PPase from A. thaliana alleviates the phenotype of the lack of the yeast m-PPase enzyme, such as bafilomycin resistance, sensitivity to alkaline pH, calcium and zinc sensitivity, and it can acidify internal components (Pérez-Castiñeira et al., 2011). This system is suitable for the heterologous expression of pathogenic organisms that encode m-PPases and for screening for inhibitors that may prove useful as therapeutic agents. Perez-Castiñeira et al. (2002b) also suggested that this system may be useful for generating cells that are tolerant to macrolides and may be further engineered to produce macrolides since they are crucial drugs in cancer and other applications.

Following the above example, Malykh et al. (2023) recently expressed the m-PPase of R. rubrum in E. coli. The result is correct since the authors generated a codon-optimized version of the enzyme (which is an exceptional tool for transferring genes from other species to bacteria and yeast cells) and found that the enzyme complements the absence of the c-PPase (the expression of the m-PPase allowed for the interruption of the c-PPase) and causes a redistribution of the carbon fluxes. The latter result is surprising and confirms that PPi hydrolysis is the ratchet that continuously flows through the synthesis of biomolecules. Nevertheless, the most surprising result is the redirection of the carbon flux from the tricarboxylic acid cycle and pentose phosphate pathways (which are ATP dependent) to an increase in acetate synthesis under aerobic growth conditions; this is correlated with a 1.5-fold increase in concentration versus the biomass generated. Thus, using m-PPase in E. coli and other biological chassis for metabolite expression may increase the production of metabolites independent of biomass growth. Additionally, this paper contributes to the study by Perez-Castiñeira et al. (2002b); even though the paper has been retracted, the authors confirm in their retraction note (doi: 10.1073/pnas.2213841119) that the conclusions are valid. They showed the complementation of S. cerevisiae expressing low levels of IPP1, the essential c-PPase. The mutant strain and plasmids are available for further experimentation,

With the findings that m-PPases are relevant for heterologous expression and have been shown to alter the phenotype of recipient cells, as a closing section, we think that an important area to explore is the use of m-PPases in plant engineering to resist drought and other stresses, with the ever-increasing threat of global warming.

First, a study by Lee et al. (2005) revealed that the metabolism of transgenic Arabidopsis plants expressing E. coli c-PPase decreased as the accumulation of glucose and fructose increased during the day (two- to threefold, 1.6- to 5.7-fold higher total soluble sugars, respectively). In contrast, starch and sucrose did not accumulate except under continuous white light exposure. Leaves in the transgenic plants presented two- to threefold accumulation of Pi and the same-fold accumulation of uridine diphosphate glucose. The metabolic effect reduced the photosynthetic activity by 20–40% but had a negligible impact on leaf size. This work revealed that a low PPi and high Pi content in the cytosol caused drastic metabolic changes, leading to increased transport in source organs (Lee et al., 2005).

The idea that PPi has a profound effect on sugar transport in plants may be relevant in plants that accumulate sugars that have important value for the ever-growing human population, according to a report by Scholz-Starke et al. (2019). An interesting feature of the H+ pump m-PPase is that it can perform the reverse reaction: PPi synthesis. The authors found a reverse mode in the flow of H+ ions by patch clamp with isolated Arabidopsis vacuoles. They confirmed the use of mutant lines lacking classical PPi-induced outward currents related to H+ pumping. Additionally, the authors used a yeast cell line expressing Arabidopsis m-PPase, and m-PPase synthesis of PPi was also observed, which is likely needed for low-oxygen respiration. The flow of H+ ions was observed only in the presence of Pi, and no reverse effect was observed. The authors also demonstrated that PPi synthesis in yeast depended on ATP in the vacuolar preparations (Scholz-Starke et al., 2019). Overall, the authors suggest that the synthetic activity of m-PPase beyond R. rubrum may impact the physiology of plants and protists. We hypothesize that this mechanism may also be true for the limited examples of m-PPases in fungi.

In an increasingly worrying global warming scenario, plant survival under stress is an important area of concern. Global warming and its impact on crops are more complex than an increase in temperature. Heat stress is linked to other environmental covariables, such as humidity, vapor pressure deficit, soil moisture content, and solar radiation (Djalovic et al., 2023). The current knowledge on the effect of m-PPase on transgenic plants suggests that PPi hydrolysis helps maintain the acidic pH of the vacuole but not reverse it. Expression of m-PPase in the apoplast induces the synthesis of glucose-1-P via UDP-glucose; glucose-1-P, in turn, is transformed into glucose-6-P and then fructose-1,6 biphosphate, which enters into respiration, and the ATP produced generates an H+ gradient that is used by sucrose synthase and then increases phloem loading (Primo et al., 2019). The accumulation of PPi, in turn, increases plant biomass and results in increased sugar accumulation, which adds to the current knowledge on plant growth. Further understanding of the growth dynamics and its control in plants is needed, including a model of PPi synthesis.

In the model A. thaliana, Nepal et al. (2020) used AVP1 transgenic plants to screen the effect of the overexpression of m-PPase on the coexpression of myoinositol oxygenase (MIOX) based on the knowledge that there are a myriad of stress conditions in which the single expression of m-PPase and MIOX results in a better response to stress. The results showed that the overexpressing A. thaliana plants exhibited better responses to salinity, drought, and temperature stress. Additionally, the crosses between AVP1 and MIOX transgenic plants yielded more seeds under water limitation stress. Importantly, the authors used a high-throughput method for assessing the chemical composition of plant tissues, applying a Scanalyzer with different sensors to capture and measure subtle phenotypes and thus obtain more information on the outcomes of the transgenic plants. Overall, the method proved useful for screening plants that produce more sugar and grow better under stressful conditions. Additionally, regarding regulation, in most instances where transgenic plants are used, m-PPase is overexpressed from the same plant, which is safe for other uses. One example is how m-PP1 from A. thaliana expressed in wheat (Triticum aestivum) induces photosynthate transport from source leaves to roots, resulting in improved yield (more seeds per plant) (Regmi et al., 2020). The same phenomenon has been reported in the moss Physcomitrella patens (Regmi, Li & Gaxiola, 2017), A. thaliana (Paez-Valencia et al., 2011) and rice (Oryza sativa) (Regmi, Zhang & Gaxiola, 2016).

The overexpression of m-PPase in plants also improved their ability to cope with nitrogen limitation. Concomitantly, the authors also investigated whether m-PPase could interact with other proteins and found that it interacts with the A. thaliana receptor-like protein kinase (AtRLK) protein via a yeast two-hybrid system and colocalization experiments. The most striking finding is that the AtRLK mutant and AtAVP1 mutant exhibit the same phenotype under low-nitrogen conditions (Zhang et al., 2023b). Overall, there are still aspects to uncover for m-PPases in plants; perhaps their roles in subcellular localization and protein interactions in plants may lead to future experiments that will help engineer plants tolerant to the increasing stress conditions in the soil, either as a direct effect of temperature or the secondary effects due to the reduction of rainfall and availability of nitrogen. The novel bacterial m-PPases capable of transporting Na+ or H+ ions may positively impact plants by enhancing their resistance to stress (Luoto et al., 2011).

Recently, vacuolar m-PPase was shown to be overexpressed in wheat plants resistant to infection by Chinese wheat mosaic virus (CWMVV) via a cell death mechanism related to increased PPi hydrolysis. Additionally, the virus produces a small interfering RNA (siRNA) that targets the 3′UTR of m-PPase, preventing the accumulation of m-PPase mRNA and thus bypassing cell death and, concomitantly, regulating H+ accumulation, leading to more favorable cellular conditions for virus replication by generating more alkaline conditions (Yang et al., 2020). The authors evaluated the effect by generating a mutant in the region that produces the siRNA, resulting in a reduction in symptoms. Thus, the enhancement of transport by m-PPase and its effect on the acidification of the vacuole are needed for viral resistance. Further studies regarding other plant hosts and viral infections could generate virus-resistant plants.

Finally, m-PPase may play an important role in resistance to bacterial infections. The Xanthomonas effector XopAP prevents stomatal closure via blockage of vacuolar m-PPase binding to phosphatidylinositol 3,5-biphosphate (Liu et al., 2022). By keeping the stomata open, the infection proceeds. Thus, m-PPase has a secondary role in bacterial infections in plants. Further research is needed to determine the safety of genetically modified plants and the use of m-PPase genes to enhance plant resistance to pathogens and stressful environmental conditions.

Conclusions

With the current knowledge of the cellular and extracellular functions of PPi, this molecule has been recognized as a fundamental bioenergetic molecule, and its hydrolysis propels subsequent biosynthetic reactions. The cumulative amount of genomic data makes PPase biology fertile for future research. The following areas can be further explored:

First, a global genomic analysis was performed to uncover novel enzymes from families II and III and the evolutionary history of PPases. One key aspect to further study is the role of the tightly bound Mn2+ ion in the currently characterized enzymes and the catalytic capacity of hydrolyzing free PPi.

Second, we aimed to determine the structure of family II enzymes with regulatory domains, particularly those from pathogenic protists, to uncover novel targets for treatment. Additionally, the identification of m-PPases in fungal organisms suggests that they may play a role in the biology of these organisms. The hypothesis presented here is that either they are involved in cell wall synthesis or they are part of an inner membrane structure and have a role as part of a proton pump to maintain an electrochemical gradient for an unknown purpose. The latter option is less likely since this enzyme was only found in oomycetes and one ascomycete. Thus, the role of m-PPase in fungi remains to be elucidated. Sequencing of environmental fungal samples will shed light on more examples of fungi encoding m-PPases. The subcellular localization in the examples found in this work may contribute to assessing their role in cell physiology.

Third, we explored the biotechnological applications of enzymes with different biochemical properties that may be linked to better RNA manipulations and other molecular biology uses.

Fourth, we aimed to assess the intracellular localization of c-PPases and determine whether they can act as protein phosphatases and regulate cellular processes, which could ultimately lead to novel targets for chemotherapy in cancer cells.

Fifth, in plant biology, m-PPases play important roles in regulating the response to stress conditions, interact with other proteins that impact plant physiology, and may be involved in resistance to viral or bacterial infections.

Finally, the importance of finding both cytoplasmic and m-PPases in pathogenic bacteria suggests that novel inhibitors may bypass the need for antibiotics or complement the inhibitory effects of currently used antibiotics. One such option is lipid-disturbing drugs.

Literature Survey and Bioinformatic Methodology

Literature survey

The literature was searched through the PubMed, Scopus and Google Scholar search engines from February to September 19, 2023. The keywords used were ‘pyrophosphatase,’ ‘inorganic pyrophosphatase,’ ‘bacteria,’ ‘membrane-bound pyrophosphatase,’ ‘proton pump pyrophosphatase,’ ‘plant pyrophosphatase,’ ‘pyrophosphatase activity detection,’ and Boolean ‘and’ for the combination of these keywords. The authors conducted an independent review of the literature to prevent any bias, and the selected articles were chosen for their relevance to the objectives of this review. Additionally, the authors independently conducted the literature review, and the number of articles addressing the subject in recent years is limited.

Bioinformatic analysis

Complementing the content of this review, sequence analysis was conducted to find new information regarding cytoplasmic and membrane-bound pyrophosphatases using the UniProt database (Release version 2023_03, June 27, 2023; The UniProt Consortium, 2023). Sequence analysis was performed using the UniProt database via BLASTp (Altschul et al., 1990) and the HMMER 2.43 online web server (release date February 2022) (Potter et al., 2018). Protein structure prediction was either retrieved from UniProt through the link to the AlphaFold Database or generated using AlphaFold2 (Jumper et al., 2021) using the collaborative environment with the default settings (Mirdita et al., 2022). Dimer prediction of cytoplasmic pyrophosphatases was performed with the AlphaFold Multimer prediction suite using the default parameters. Phylogenetic analysis was conducted with MEGA version 11.0.13 (Tamura, Stecher & Kumar, 2021). The phylogenetic analysis parameters are indicated in each figure legend. The groups identified were manually colored.

Protein structure alignment was conducted using the default settings with the US-align web server (Zhang et al., 2022).

Supplemental Information

Supplemental Information 1 All models and sequences used in this study

Supplemental Information 2 Supplementary Figures (1-4)

Supplementary Figure 1. Clostridium tetani encodes for both a family II CBS-domain enzyme and a m-PPase. (A), UniProt entry A0A4Q0VED8 corresponds to a family II PPase; the model is shown in rainbow color. (B) compares the C. tetani enzyme with the B. subtilis family II PPase (RMSD = 2.43, TM-score 0.462). (C) is the model for the predicted K+ stimulated m-PPase of C. tetani (UniProt A0A4Q0V5P6), rainbow color scheme. Supplementary Figure 2. S. carlsbergensis (Saccharomyces pastorianus) cytoplasmic PPase is inconsistent with a membrane-bound enzyme. UniProt entry A0A6C1DXI6 encodes for a second PPase (the family I enzyme consistent with other family I enzymes is A0A6C1DM74, and an almost identical copy, UniProt A0A6C1E2Z3). The model shows that the N-terminal domain is also a helix (as for E. histolytica) with extensive negative charge. This suggests that the enzyme may not be associated with the membrane but co-purified along with a cellular membrane. Supplementary Figure 3. Protter analysis of the selected m-PPases is shown in Figs. 10 and 12. The number indicates the position of each cluster of enzymes in Fig. 10. Asterix indicates the Protter prediction for R. rubrum enzyme, and the model suggests the position of the catalytic residues (in red). The upper part of each figure corresponds to the outside of the membrane, and the lower part to the cytoplasmic side. Supplementary Figure 4. The finding of an OmpA-like domain in a m-PPase. In search of the diversity of m-PPases and group I of Fig. 10, we found C-terminal extensions in the examples analyzed; by searching in UniProt by protein length, we found that in Paucibacter KCTC 42545, the m-PPase (UniProt A0A0U2VTE8) was 810. In (A), the AlphaFold2 model shows that the C-terminal end contains an extra domain. In (B), HMMER analysis showed that the extra domain exhibits similarity to OmpA protein, and the sequence is conserved among bacteria, as examples: Burkholderiales bacterium, Rubrivirax sp., Rhizobacter, Acidovorax sp., Polaromonas sp. The domain shows variable conservation in the extension of the sequence (upper graph), and the distribution is shown to be in bacteria exclusively (middle and lower images). In (C), a comparison of the full-length model of Escherichia coli K-12 OmpA (UniProt P0A910) is presented in a rainbow color scheme. Also, the structural alignment shown in (D) (RMSD = 5.88, TM-score = 0.052), although very low, suggests that the extra domain in the m-PPase (in blue) is a remote resemblance to OmpA (in red).

We dedicate this manuscript to the loving memory of Professor Heliodoro Celis, Institute for Cellular Physiology, UNAM, the greatest mentor and scientist who promoted research on pyrophosphatases of nonsulfur purple photosynthetic bacteria.

Additional Information and Declarations

Competing Interests

Author Contributions

Data Availability

Rodolfo García-Contreras, Héctor Manuel Mora-Montes and Bernardo Franco are Academic Editors at PeerJ. The other authors declare that they have no competing interests.

Rodolfo García-Contreras conceived and designed the experiments, analyzed the data, prepared figures and/or tables, authored or reviewed drafts of the article, and approved the final draft.

Javier de la Mora performed the experiments, authored or reviewed drafts of the article, and approved the final draft.

Héctor Manuel Mora-Montes performed the experiments, prepared figures and/or tables, authored or reviewed drafts of the article, and approved the final draft.

José A. Martínez-Álvarez performed the experiments, authored or reviewed drafts of the article, and approved the final draft.

Marcos Vicente-Gómez performed the experiments, authored or reviewed drafts of the article, and approved the final draft.

Felipe Padilla-Vaca performed the experiments, prepared figures and/or tables, authored or reviewed drafts of the article, and approved the final draft.

Naurú Idalia Vargas-Maya conceived and designed the experiments, analyzed the data, prepared figures and/or tables, authored or reviewed drafts of the article, and approved the final draft.

Bernardo Franco conceived and designed the experiments, analyzed the data, prepared figures and/or tables, authored or reviewed drafts of the article, and approved the final draft.

The following information was supplied regarding data availability:

The models and sequences are available in the Supplementary File.

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
