# Peer review of "The inorganic pyrophosphatases of microorganisms: a structural and functional review"

_PeerJ, doi:10.7717/peerj.17496_

## Round 0.1 · original submission · Major Revisions

· Academic Editor

Major Revisions

Please revise the manuscript by following the reviewers' comments and suggestions. A detailed point-by-point response is required when re-submitting the revised manuscript. Please address all the comments and suggestions carefully and thoroughly.

**Language Note:** The review process has identified that the English language must be improved. PeerJ can provide language editing services - please contact us at [email protected] for pricing (be sure to provide your manuscript number and title). Alternatively, you should make your own arrangements to improve the language quality and provide details in your response letter. – PeerJ Staff

Reviewer 1 ·

Basic reporting

The review by Contreras et al. focuses on microbial pyrophosphatases - their structures, distribution, function, and practical uses. The authors consider PPases to be understudied enzymes and suggest the direction of their future studies that would allow a deeper understanding of this enzyme and its use for practical purposes. Therefore, they put emphasis on PPases “with oddities” and on the aspects which have not yet been studied with scrutiny. The review combines in one place a lot of useful information and will be of use for a wide audience. Naturally, the review does not equally cover all aspects of PPases, for instance mechanism of catalysis.
The review is generally well written, but some clarification and English editing are needed as described below.

Lines 115-21: Actually, the scheme described refers to any PPase. not only to family II PPases.

Line 131 and elsewhere: It is the accepted practice to use one-letter amino acid codes in sequences only.

Line 187: Conversion of nucleophilic water into reactive hydroxide is very likely in families I and II, wherein the water is coordinated by several metal ions, but not in mPPase, wherein it is coordinated by two carboxylates. They can only further polarize it to increase nucleophilicity, like in pepsin.

Line 264, etc: This is a pure hypothesis, not a proven fact, as is presented here.

Lines 277: This contradicts the view that families I and II are nonhomologous proteins.

Line 408: It is not exactly correct to state that family II PPases are most active with Mn2+. Under physiological conditions, this enzyme has a very tightly bound Mn2+ or Co2+ ion, which very slowly exchange with other metals. But they additionally require Mg2+ ions for full activity, which act together with Mn/Co in catalysis.

Line 411: The author should refer to Huang et al. (Biochemistry 2011, 50, 8937-8949) wrt the structure of family III PPase.

Line 430: Family I PPases are mostly hexameric in prokaryotes but dimeric in eukaryotes.

Line 439: “activated water molecule”.

Line 469: “The role of Co2+ or Mn2+ in the family II enzymes has been controversial.” What does the controversy consist in?

Lines 488-494: The meaning of this long sentence is unclear.

Lines 527-530: The two sentences apparently contradict each other.

Line 562: A more recent publication (Zamakhov et al. Int. J. Mol. Sci. 2023, 24, 17160) describes the architecture of tetrameric CBS-domain-containing family II PPase, based on cryo-EM and modelling data.

Line 663: plant vacuolar mPPase pumps into cytoplasm.

Lines 664 and 671: The correct spelling is Nordbo.

Line 674-675: Sentence meaning is unclear.

Line 685: it is incorrect to call alpha-helices as domains. In the generally accepted meaning of “domain” all mPPases are single-domain proteins.

Lines 771-778: This text partly repeats some text above.

Line 840: “PPi is not an energy-driving force” is unclear.

Experimental design

no comment

Validity of the findings

no comment

·

Basic reporting

"The inorganic pyrophosphatases of microorganism: a structural and functional review" is a nice review which almost summarizes all the reaserch progress of inorganic pyrophosphatases, focusing on structural and functional of this kind enzyme. The literature references are sufficient and the figures are clear.

Experimental design

Authors predicted several structures of inorganic pyrophosphatases by using alphafold. The results are clear and enough. Authors also classified the enzyme based on amino acid sequence and 3-D structures. They also enough people to continue study PPase because there are still some unresolved mysteria among this kind enzyme.

Validity of the findings

This review is an important contribution to PPase study community.

Reviewer 3 ·

Basic reporting

The manuscript “The inorganic pyrophosphatases of microorganism: a structural and functional review” by R. García-Contreras et al. describes various aspects of pyrophosphatase (PPase) structure, function and applications. The information on these enzymes is of broad interest for molecular biologists, biochemists, bioenergetics etc., and the field reviewed here clearly falls within scope of the journal. The review covers a wide spectrum of PPase-related topics, from structural and functional aspects to phylogeny and the role of pyrophosphate to biotechnology applications to pyrophosphate detection, and includes 156 literature sources. Such a large review encompassing all those different aspects was not previously published in this field, although reviews on some separate topics were published in recent years. The authors made a large work, both reviewing literature data and performing their own experiment. However, I have serious critical comments to this manuscript.

Unfortunately, the large material is not well structured, which is a weakest point of this manuscript. Section titles do not correspond to their content. Introduction, instead of introducing a reader into the problem and authors’ motivation, is too large and presents a lot of major information which would be more appropriate in the main body of a review (e.g., review on the catalytic mechanism of PPases). Main body of the text does not have a title. Most of the sections lack inner structure of lower-level sections. A section “PPases and applications”, among applications per se, lists other topics, for instance, novel approaches to PPi quantification and search for novel PPase inhibitors, without separation between these topics. Arrangement of material in these sections lacks a system, there are repetitions and logical inconsistencies. For instance, a question on PPi synthesis against hydrolysis is repeatedly and at length discussed: in Introduction (line 61 and further) and in the section Membrane-bound PPases (line 726 and further). Similarly, a question on PPase diversity and phylogeny is discussed first in Introduction (line 166 and further) and then again in line 872. I suggest that the material should be divided into proper sections and sub-sections corresponding to the content, with logical argumentation of some points and without repetitions.

Experimental design

In addition to the lack of logical organization, and probably due to it, the second problem of this manuscript is the incomplete representation of material in some sections. For instance, when reviewing soluble PPases, the authors focus on cytoplasmic PPases, while bacterial homologs or plastid isoforms of eukaryotic PPases are only occasionally mentioned. Meanwhile, catalytic mechanism and structural features of bacterial enzymes are well represented in the literature. In the discussion of novel inhibitors of soluble PPases, the authors only mention fructose-1-phosphate as inhibitor of E.coli PPase (apart from fluoride or substrate analogues), although a number of novel synthetic and natural inhibitors described for M.tuberculosis PPase (see e.g. Pang 2016) are not mentioned. Another example is discussion of the role of Gly residues in catalysis (lines 146-152), while the role of other residues and corresponding studies are not discussed. The authors say in the Survey Methodology that “selected articles were chosen for their relevance to this topic” (line 299), so this incomplete coverage may be the result of wrong methodology with some kind of bias.

The incomplete analysis of discussed problems leads to incorrect statements. For instance, in the description of a catalytic mechanism by Family I PPases, a low-barrier H-bond between protein residue and an attacking water molecule is listed as the only factor “to provide the catalytic power to the enzyme” (line 144). The role of metal ions in catalysis is not described at all, hence the text yields a wrong impression about the catalytic mechanism. Another example of incorrect statement is the description of the oligomeric forms of PPases: (lines 156-158) “In the case of family I enzymes, it has been found to form homohexamers (the native form of the enzyme), homotrimers, or homodimers, depending on specific mutations on the protein” – this statement can be applied to bacterial Family I PPases, while eukaryotic representatives are usually dimeric with several exceptions. The statement that m-PPase "needs the ion gradient" (line 89) is taken out of context, it is only correct for the PPi synthesis direction, but this important point is not mentioned in the text. Discussing the differences in Family I, II and III PPases (116-119), the authors make a focus on the MgPPi binding and the number of constants along the catalytic cycle, while there are other, more important differences; e.g., Family III PPases are in fact haloalkane dehalogenases. Discussing the role of PPi in the cell, the authors say that, among "402 reactions of the universal biosynthetic core; none of the reactions use PPi", which is again incorrect, since there are well-known reactions and enzymes using PPi as a source of phosphate group and energy, e.g. PPi-dependent kinases. These mistakes are clearly the result of superficial analysis or careless usage of literature sources.

In addition to the review part of the manuscript, the authors also performed the experiment involving bioinformatic search and structural modelling. This work presents some interesting results. However, these sections are not properly organized, e.g. experimental data are not separated from the literature data. As a consequence, it is difficult to evaluate the experimental results. Probably the authors would consider to describe this material in the separate section, with the proper description of Experimental methods and Results. Computational experiment is not a “survey”, it should be properly performed, analyzed, and presented. These results should be discussed in view of (and in combination with) data from literature, with clear indication, where is what. In the current variant, there is a mix of these two, both in the text and in the figures.

Validity of the findings

The Introduction does not set up any clear goals. Literature review part of the text fails to provide logical arguments, it mostly contains poorly structured pieces of information. The part involving authors' own results contains adequate arguments. However, their motivation and possible importance is unclear.

Conclusions contain interesting points concerning authors' vision of possible future evolution of PPase study. However, as well as Introduction, it includes some material that should be placed in the main body of the text.

Additional comments

The English language needs to be improved. In a current variant, it is difficult at places to understand the authors’ ideas.

Reviewer 4 ·

Basic reporting

The review article aims to summarize the current literature on inorganic pyrophosphates (Ppases), analyzes all Ppases available so far in uniprot, and makes a strong case on why continued research on this enzyme is essential.

This is an important piece of work, but I found it hard to keep up with the article without getting lost. I believe it would heavily benefit from restructuring to ensure the article retains the flow and connectivity between the sections. As the readers of this review will range from individuals with varying level of expertise about this enzyme, it is essential for the article to be relatively easy to follow and understand in order to achieve its goal.

Experimental design

* When was the literature survey performed? Dates are missing.
* Version info is missing for Uniprot database used. Similary, version info was missing for tools such as BLASTp and US-align. Also, the dates info on when HMMER server was used is missing.
* Sentence in line 301 is incomplete.("Also, the literature review was conducted independently by")

Validity of the findings

NA

Additional comments

Amino acid and their position are notated inconsistently. For example, C16 in lines 341 and 344; line 341 uses superscript and line 344 does not. Similar inconsistencies elsewhere as well (line 141, for example). Personally I would suggest usage of three letter code for amino acids due to their readability (i.e. Cys16 instead of C16).

---

## Round 0.2 · Major Revisions

· Academic Editor

Major Revisions

Please revise the manuscript based on the suggestions of the reviewers. When re-submitting your revised manuscript for consideration, please also provide a point-by-point response letter to show that all the issues raised by the reviewers have been well addressed.

**Language Note:** The review process has identified that the English language must be improved. PeerJ can provide language editing services - please contact us at [email protected] for pricing (be sure to provide your manuscript number and title). Alternatively, you should make your own arrangements to improve the language quality and provide details in your response letter. – PeerJ Staff

Reviewer 1 ·

Basic reporting

I find that the manuscript has improved considerably after the revision. Yet it still needs minor language refinement. Some examples where the language could be improved are given below. I suggest contacting a professional editing service or a person who is proficient in English and familiar with the subject matter.
Lines 144-145: “which is located in the active site by concentration from Trp100 hydrophobicity and Asn116” – the meaning is unclear.
Lines 155-157: the sentence needs recasting.
Lines 219-222: This is a pure hypothesis, and this should be clearly indicated.
Lines 695-696: “enzymes linked to energy couplers” – unclear meaning.
Some paper titles are overcapitalized in the Literature section.

Experimental design

n/a

Validity of the findings

n/a

Additional comments

n/a

Reviewer 4 ·

Basic reporting

I appreciate the authors for making many of the requested changes. However, my major concerns regarding poor structuring of the manuscript and lack of continuity between and within the sections continue to be present in this revision as well.

* Structuring/organization of the manuscript needs serious reconsideration. Currently, it is really hard to follow the manuscript due to poor organization as well as lack of continuity. For example, section "Do m-PPases are present in fungi?" is detailed from line 238, but section "Membrane-bound PPases" are described way latter from line 660.
* Same concept gets introduced multiple times, and this results in confusion instead of reinforcing the concept. For example, classification of c-PPases into family I and II gets introduced multiple times at lines 120, 329, 431, etc.
* In addition to literature review, authors performed their own survey on all available PPase sequences in UniProt. However, introduction section does not sufficiently describe this review+own survey setup. As methodology section has been moved to the end, it further adds to the confusion and readers would find it hard to understand the review.
* It is often unclear which claims are part of the literature review and which are from the survey/analyses performed as part of the study in this manuscript. Please be sure they can be differentiated easily and clearly.
* Citations are sometimes incomplete. As this is a review, citations take additional importance. For example, lines 248-251, 330-332, etc. are missing citations.
* As the manuscript discusses c-PPases as well as m-PPases, it is important that they are always explicitly identified as such. In several instances, they are simply mentioned as PPases which can lead to confusion. For example, see lines 307, 402, 479, etc.
* On the similar note, please be sure that c- and m-PPases are used properly. For example, line 199 says c-PPase, when it should be m-PPase.
* Phrases "in the following sections", "in the following lines" and "in previous lines" used in the manuscript would benefit from being explicit about the sections they are referring to. As the manuscript is quite long, it is not clear what the authors are referring to.

Experimental design

NA

Validity of the findings

NA

Additional comments

NA

---

## Round 0.3 · Minor Revisions

· Academic Editor

Minor Revisions

Please follow the reviewer's final comments to further revise your manuscript.

Reviewer 1 ·

Basic reporting

no comment

Experimental design

no comment

Validity of the findings

no comment

Additional comments

I find that text quality, which was the main problem of the original manuscript, has increased markedly, making this interesting and timely survey publishable in PeerJ. To finally polish the manuscript, please consider some minor remarks shown below.

Lines 56, 61, 95, 609: the correct full term “(trans)membrane gradient of the electrochemical potential of an ion” should not be сut to “electrochemical gradient”. This is electrochemical potential gradient, which should be related to the specific molecule or ion.
Line 109: looks irrelevant.
Line 124: reversibility sign is replaced.
Lines 130-131: consider replacing “the constants for PPi hydrolysis proceed faster than those for product removal” with “the PPi hydrolysis proceeds faster than product removal”.
Line 261: Correct author’s name to “Salminen” throughout the manuscript.
Line 290: replace “activated” by “activation”.
Line 481: …extensions with CBS and DRTGG domains. (?)

---

## Round 0.4 · accepted · Accept

· Academic Editor

Accept

The authors have revised the manuscript thoroughly based on all the reviewers' comments.